# Regional cooling caused recent New Zealand glacier advances in a period of global warming

Andrew N. Mackintosh[1,2,*], Brian M. Anderson[1,*], Andrew M. Lorrey[3,*], James A. Renwick[2], Prisco Frei[1] & Sam M. Dean[4]

Glaciers experienced worldwide retreat during the twentieth and early twenty first centuries, and the negative trend in global glacier mass balance since the early 1990s is predominantly a response to anthropogenic climate warming. The exceptional terminus advance of some glaciers during recent global warming is thought to relate to locally specific climate conditions, such as increased precipitation. In New Zealand, at least 58 glaciers advanced between 1983 and 2008, and Franz Josef and Fox glaciers advanced nearly continuously during this time. Here we show that the glacier advance phase resulted predominantly from discrete periods of reduced air temperature, rather than increased precipitation. The lower temperatures were associated with anomalous southerly winds and low sea surface temperature in the Tasman Sea region. These conditions result from variability in the structure of the extratropical atmospheric circulation over the South Pacific. While this sequence of climate variability and its effect on New Zealand glaciers is unusual on a global scale, it remains consistent with a climate system that is being modified by humans.

[1] Antarctic Research Centre, Victoria University of Wellington, Wellington 6140, New Zealand. [2] School of Geography, Environment and Earth Sciences, Victoria University of Wellington, Wellington 6140, New Zealand. [3] National Institute of Water and Atmospheric Research, Auckland 1010, New Zealand. [4] National Institute of Water and Atmospheric Research, Wellington 6021, New Zealand. * These authors contributed equally to this work. Correspondence and requests for materials should be addressed to A.N.M. (email: Andrew.Mackintosh@vuw.ac.nz).

Glaciers are iconic indicators of climate change, and understanding their anthropogenic and natural drivers is critical for anticipating future sea-level rise and freshwater security. Monitoring of terminus positions for ~500 glaciers worldwide has revealed a largely homogeneous trend of retreat during the late twentieth and early twenty first centuries[1]. Unprecedented global ice loss has occurred during the last three decades, which are the warmest of the instrumental era[2]. The predominantly negative global glacier mass balance between 1991 and 2010 has been primarily attributed to anthropogenically-forced warming[3]. Examples of glacier stability and expansion during this period that are attributable to climate, rather than to internal glacier dynamics, are confined to just a few regions such as the Karakoram and Pamir[4–6], western Scandinavia[6,7], southern Patagonia[6] and the Southern Alps of New Zealand[6,7].

A number of glaciers in New Zealand recently exhibited a globally exceptional period of terminus advance (Figs 1 and 2). The World Glacier Monitoring Service database shows that 58 glaciers advanced at some point in the 1980s, 1990s and early 2000s (refs 8,9), and 12 of these glaciers advanced continuously for five or more years. In 2005, when this glacier advance phase neared its end, 15 of the 26 advancing glaciers observed worldwide were in New Zealand[9]. Franz Josef, the glacier with the most complete length change record in the Southern Hemisphere, advanced for 19 of the 25 years between 1983 and 2008, regaining almost half (1.4 km) of the ~3 km length it lost between 1893 and 1982 (ref. 8; Fig. 2). We use the length record at Franz Josef Glacier to define the period of climate forcing associated with this glacier advance phase. By taking into account the ~3-year time lag between climate forcing and terminus response observed at Franz Josef Glacier[8], we suggest that the corresponding climate period extended from 1980 to 2005.

Of the New Zealand glaciers that recently advanced, Fox and Franz Josef glaciers are the best-monitored (Fig. 2)[8]. These adjacent, steeply inclined glaciers transport very large amounts of snow and ice (accumulation rates of ~10 m per year water equivalent, and ice flow rates of up to 1 km per year) through deep, narrow valleys to elevations only a few hundred metres above the sea level. At such elevations, these glaciers sustain the highest surface melt rates on Earth (~20 m per year water equivalent)[10]. This combination of large mass balance gradients and steep, relatively thin and fast-moving ice makes them adjust rapidly to climate forcing. Very few glaciers on Earth are capable of responding this quickly.

The response of gently sloping glaciers in the Southern Alps contrasts with the steep glaciers described above. The difference is most stark for the low-angle, debris-covered glaciers that terminate in lakes, which have generally down-wasted rather than retreated horizontally, until pro-glacial lake development recently began[11,12]. No Southern Alps glaciers with significant terminal lakes have advanced in the last four decades. Such glaciers are in decline; for example, Tasman Glacier, New Zealand's largest, has undergone ~5 km of retreat into a terminal lake since the early 1980s (refs 13,14) (Fig. 2). Because Tasman and other lake-calving glaciers constitute approximately half of New Zealand's total glacier volume, Southern Alps ice volume as a whole has decreased since the 1980s (ref. 15), even though several glaciers have advanced during this period.

Although it has been debated whether recent New Zealand glacier fluctuations relate more to temperature or precipitation forcing (for example, refs 16,17), recent studies have suggested that increased precipitation was mostly responsible for the glacier advance phase between 1980 and 2005 (refs 7,18). The Intergovernmental Panel on Climate Change Fifth Assessment Report summarized these arguments, stating 'The exceptional terminus advances of a few individual glaciers in Scandinavia and New Zealand in the 1990s may be related to locally specific climatic conditions such as increased winter precipitation'[6]. However, this summation remains speculative because direct glacier mass balance and high-elevation climate data from the Southern Alps are limited. Furthermore, attributing this particular glacier advance phase to a climate driver requires a physics-based approach, which allows linkages between glacier changes and the Southern Hemisphere climate system to be fully examined.

Glacier mass balance depends mainly on air temperature, solar radiation and precipitation. Meteorological campaigns on glaciers

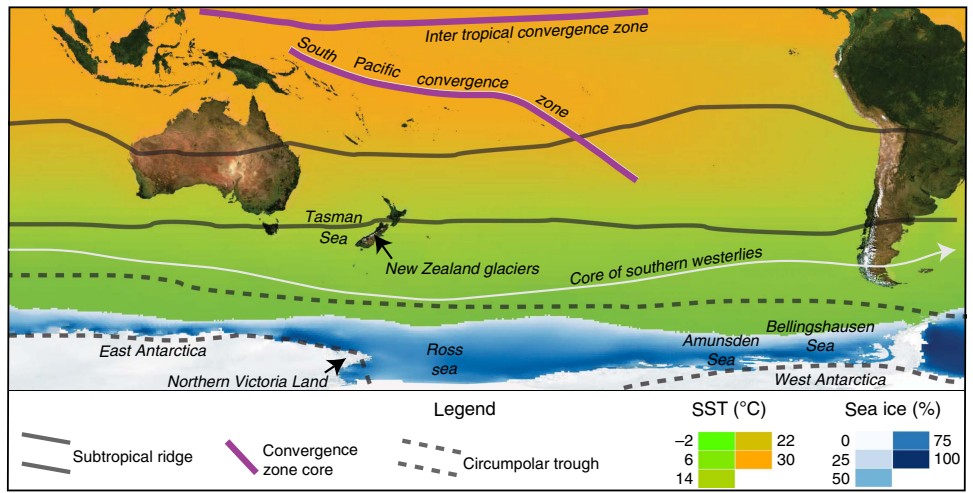

**Figure 1 | Southern Hemisphere climatological features and New Zealand glaciers.** Most New Zealand glaciers are located in the Southern Alps, a mountain range extending for >500 km from north to south along the South Island of New Zealand. The Southern Alps are orientated perpendicular to the prevailing mid-latitude westerly winds and rise to ~3,000 m. The climate of the South Island is influenced by processes to both the north and the south. To the north lies the subtropical ridge, South Pacific Convergence Zone and Inter Tropical Convergence Zone. To the south lies the core of the westerlies, the circumpolar trough and Antarctic sea ice. All of these features have the potential to influence atmospheric and oceanic conditions in the New Zealand region, and hence Southern Alps glacier mass balance. Sea surface temperature (SST) is from annual mean data, while sea ice data show peak concentration (%) reached in the austral spring. Other climatological features are plotted in their mean annual positions.

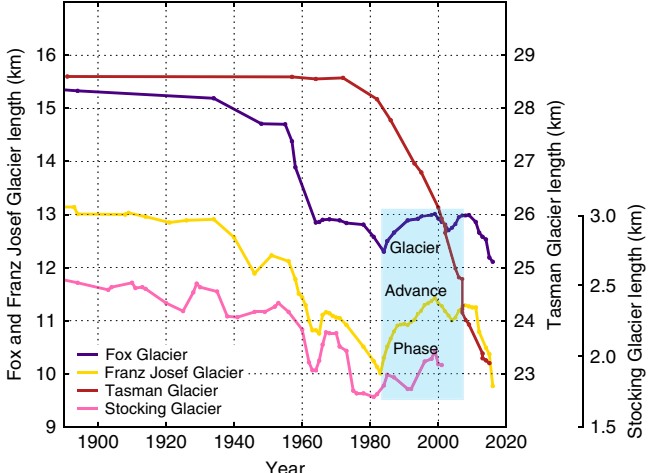

**Figure 2 | Historic length changes for four glaciers in New Zealand.**
Franz, Fox, Stocking and Tasman glaciers (see Fig. 3 for glacier locations) retreated during the twentieth and early twenty first centuries. However, Franz Josef, Fox and Stocking glaciers also experienced periodic re-advances. In this paper we identify the climatological and glaciological drivers of the largest and most recent of these re-advances between 1983 and 2008 (marked by blue shading). The three glaciers that advanced are all steeply inclined and react swiftly and similarly to climate forcing. Fox and Franz Josef glaciers, both >10 km long, flow to the west and north from the major drainage divide (main divide) of the Southern Alps. Stocking Glacier is much shorter than Fox and Franz Josef glaciers, and flows to the east of the main divide. Tasman Glacier has a gentle slope and is the largest and thickest glacier in New Zealand. During the twentieth century, Tasman Glacier experienced continuous thinning, followed by ~5 km of retreat via pro-glacial lake formation since the 1980s.

have shown that the primary source for melt energy is solar radiation, but that year-to-year fluctuations in mass balance are mainly due to temperature and precipitation[19]. Maritime glaciers including those in the Southern Alps of New Zealand (Fig. 1) are particularly sensitive to temperature change via its influence on turbulent heat fluxes, long-wave radiation and the elevation of the snow/rain threshold[19–23]. Consequently, precipitation increases must be substantial to offset warming[19,24]. An energy balance approach to glacier mass balance modelling accounts for the aforementioned variables and interactions between them (for example, ref. 25), and has the potential to identify and rank the most important climatic drivers of glacier changes.

Here we use a regional-scale energy balance model to test the hypothesis that New Zealand glacier advances between 1983 and 2008 resulted from precipitation increase. We also examine the alternative hypothesis that regional cooling was responsible. We show that the primary hypothesis is not supported; advance of glaciers in the Southern Alps between 1983 and 2008 was mostly due to reduced air temperature rather than increased precipitation. The lower air temperatures in the mid-1980s, early to mid-1990s and mid-2000s are linked to variability in the extratropical atmospheric circulation. In particular, the influence of large-scale atmospheric waves (Pacific South American and Zonal Wave 3 patterns[26]) encouraged southerly winds in the New Zealand region, cooling the Tasman Sea and lowering air temperature in the adjacent Southern Alps. For some New Zealand glaciers, this combination drove glacier advances that ran counter to the global trend.

## Results

**Simulated and observed glacier mass balance.** We simulated glacier mass balance between 1972 and 2011 on a model domain

covering the central Southern Alps, a region that contains approximately one-third of New Zealand's ice-covered area, and two-thirds of its ice volume[27] (Fig. 3, Supplementary Fig. 1). In our 'standard run', we forced a glacier mass balance model with gridded climate data[28,29] and re-analysis data sets[30], using a standard set of model parameters (See Methods and Supplementary Table 1). This simulation captured distinctive aspects of glacier mass balance in the Southern Alps, including accumulation rates of up to 10 m and melt rates of to 20 m per year water equivalent at Franz Josef Glacier[31] (Fig. 3). While there are no glacier-wide mass balance observations available within the model domain, our standard run demonstrates skill in replicating direct point mass balance observations collected from a number of glaciers over this 39-year period (see Methods, Supplementary Fig. 2, Supplementary Table 2). Over this time period, we find that 49% of the total surface energy comes from turbulent fluxes, 41% from net radiation and 10% from precipitation. The proportion of energy coming from precipitation is higher than found by other studies in maritime environments (for example, ref. 32), but is not surprising, given the high rainfall rates (up to 12 m per year) and warm temperatures within our model domain, especially on the many glaciers that reach low elevations to the north and west of the main divide (Fig. 3).

Figure 4a shows simulated annual mass balance for the standard run for the largest glaciers in our domain. This simulation shows a large degree of interannual variability in glacier mass balance. Particularly negative mass balance years are simulated in 1990, 1999 and 2000. However, sustained periods of positive mass balance are also simulated, in particular during the mid-1980s, in the early to mid-1990s and in the mid-2000s. New Zealand lacks long-term measurements of glacier mass balance to compare directly to this simulation. However, annual snowlines on a number of Southern Alps glaciers (Supplementary Fig. 1 and Supplementary Table 3) have been surveyed by oblique aerial photography at the end of each glacier year (austral summer, typically in March) since 1976 (refs 15,33). Our simulated glacier mass balance anomalies and these recently measured annual snowlines are strongly positively correlated ($r = 0.69$, $P < 0.01$), suggesting that the modelled year-to-year variability is realistic (Fig. 5a).

Simulated glacier mass balance varies significantly between glaciers within our model domain. Our modelling suggests that large, low-angle glaciers such as Tasman and Murchison (Fig. 3) experienced mostly negative mass balance and overall ice volume loss during this period (Fig. 4). This is consistent with observations that these glaciers have been losing volume between 1972 and 2011 (ref. 15). In contrast, Fox and Franz Josef Glaciers show overall positive glacier mass balance, and gain volume in this 39-year simulation, also consistent with observations[8] (Fig. 4). Because glacier length has been monitored closely at Franz Josef Glacier, and also because it can respond to climate variability within a few years[8], the record of terminus fluctuations at Franz Josef provides a key target for comparison with our simulated glacier volume.

Figure 5b shows that the modelled glacier volume increased at Franz Josef Glacier almost continuously from 1983 to 1999, and again between 2005 and 2008. The model also simulates a period of mass loss in the late 1990s and early 2000s that corresponded with observed terminus retreat. There is a 2–3 year time lag between glacier volume changes simulated by the model, and length changes recorded at Franz Josef Glacier. This time lag is similar to the 3–4 year terminus 'reaction time' estimated for Franz Josef Glacier based on terminus observations[8]. Overall, these results provide confidence that our mass balance model is simulating the major processes, both in space and time, that

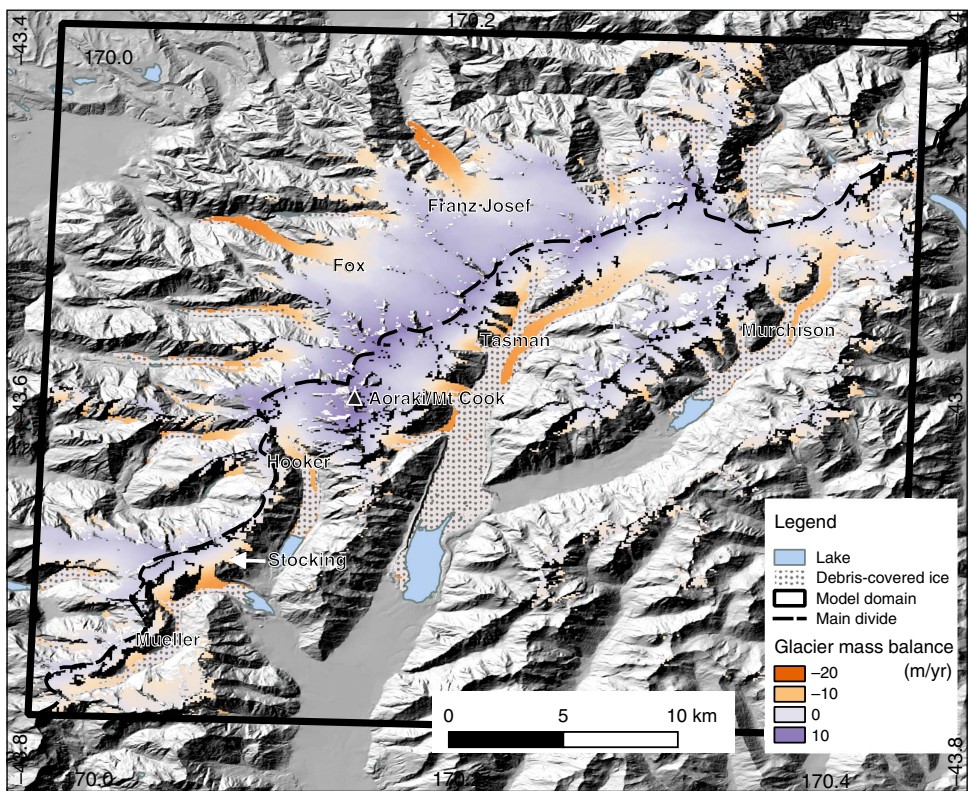

**Figure 3 | Model domain in the central Southern Alps of New Zealand.** The map shows the location of major glaciers, mean glacier mass balance (1972–2011), surface debris cover and pro-glacial lakes. The Southern Alps contain more than 3,000 glaciers, but the greatest volume of glacier ice in New Zealand is located within our model domain at 43°S, approximately centred on Aoraki/Mt Cook, New Zealand's highest mountain (3,724 m). Mass balance (metres of water equivalent per year) is shown in red (net melt) and blue colouring (net accumulation). Very large gradients in glacier mass balance exist within the model domain, depending on glacier elevation and location with respect to the major drainage divide (main divide). Franz Josef and Fox glaciers to the west and north of the main divide each show snow accumulation rates of ∼10 m, and melt rates of ∼20 m of water equivalent per year. Surface debris covers the lower portion of many glaciers including the Tasman, Hooker, Mueller and Murchison glaciers. Terminal lakes have grown rapidly at these glaciers since the 1980s.

forced New Zealand glacier mass balance changes between 1972 and 2011.

**Diagnostic experiments with energy balance model.** To evaluate whether precipitation, temperature or another climate variable caused the glacier advance phase between 1980 and 2005, we carry out a series of diagnostic experiments. First, we create synthetic data sets for each climate variable where the value for each day has an offset added so that the mean for each year is constant across the simulation at the long-term mean value. Second, we carry out experiments where only one climate variable is allowed to vary as observed, while our synthetic climate series are used for the other model inputs. The glacier volume change resulting from each run is then taken as the contribution of that variable to the overall volume change (Fig. 6c,d).

The experimental set-up involves three different types of model run. The standard run (Figs 4 and 6) is used to assess how glacier mass balance has changed overall, and is a simple, unadjusted model simulation from 1972 to 2011. The second is called the 'control run'; in this case, the model is run with all climate variables adjusted so that, for each year, the annual mean is the same as the long-term mean and daily variations are retained. In the third run type, the variable of interest is allowed to vary without modification, while the other variables are held at their long-term climatological means (again at an annual

resolution, with daily variations retained). The results are presented for each variable as anomalies from the control run (each named 'temperature run', 'precipitation run' and so on). The output of this third type of run is taken as the contribution of that variable to the cumulative glacier volume change (Fig. 6d). Such an interpretation is reasonable because the main climate variables affecting the model (temperature and precipitation) are uncorrelated ($r = 0.12$, $P > 0.1$; Supplementary Fig. 3). Furthermore, the individual model runs described below sum almost exactly to the standard run, which includes all forcings, suggesting very little interaction between these climate variables.

To test the hypothesis that precipitation increase caused the glacier advance phase, we assess the relative contribution of precipitation variability (Fig. 6a) to glacier mass balance by holding all other climatic variables at their mean values. This experiment illustrates that from 1972 to 1979, precipitation had a negative influence on mass balance (resulting in volume loss), while between 1979 and ∼2000, it had a positive influence (Fig. 6d). Although this is consistent with the hypothesis that increased precipitation caused the glacier advance phase in New Zealand, our diagnostics indicate that precipitation variability accounts for only 27% of the total ice volume anomaly during the advance phase (Fig. 6c).

We repeated the experiment to examine the influence of temperature variability (Fig. 6b) on mass balance changes by holding the other climate variables at their mean annual values. The resulting pattern (Fig. 6d) shows that temperature change is

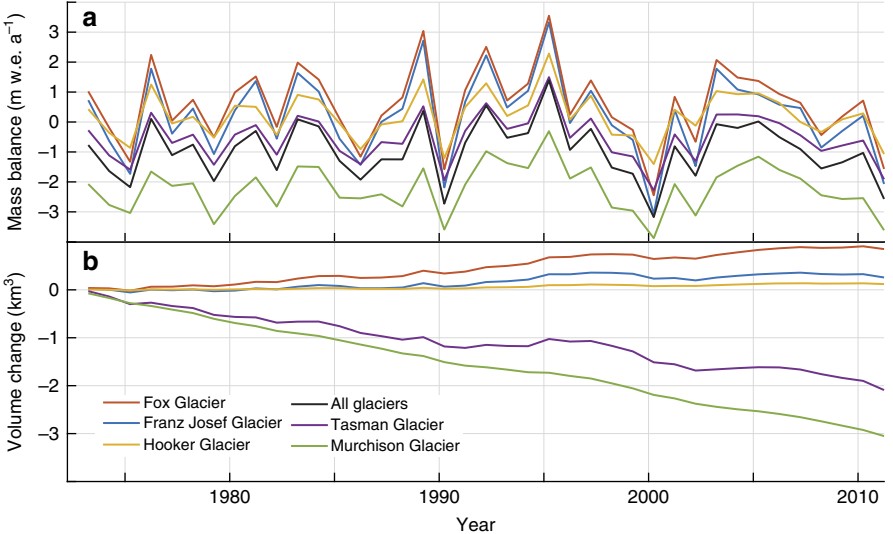

**Figure 4 | Simulated mass balance and ice volume changes.** (**a**) Simulated mass balance and (**b**) simulated ice volume change in the central Southern Alps between 1972 and 2011 for the five largest glaciers within our model domain (from the 'standard run'). Mass balance units are metres of water equivalent per year (m w.e.a $^{-1}$). (**a**) All glaciers show significant interannual variability in glacier mass balance, with positive years in the mid-1980s, early to mid-1990s and mid-2000s. Mass balance at Fox, Franz Josef and Hooker glaciers was more positive overall, while mass balance of Tasman Glacier remained negative except for a few years in the 1990s. Murchison Glacier mass balance stayed negative for the entire period. Note that the mean mass balance for all glaciers (black line) is identical to that shown in Fig. 9 (standard run, all glaciers). (**b**) The lower panel shows how the cumulative effect of mass balance fluctuations results in glacier volume change for five glaciers in our model domain. Fox and Franz Josef glaciers gained volume overall, consistent with observations that they advanced between 1983 and 2008. Tasman and Murchison glaciers lost volume, again consistent with observations.

the dominant variable that caused the glacier changes, accounting for 56% of the total volume anomaly during the glacier advance phase (Fig. 6c). The majority of this temperature-driven volume increase occurred during two periods, in the 1980s and again during the 1990s (Fig. 6b). The experiment was repeated with wind, cloudiness and relative humidity, but the combined effect of each of these variables resulted in small (17%) changes in glacier volume, compared with temperature (56%) and precipitation (27%; Fig. 6c). While these exact percentage contributions of different climate drivers vary slightly depending on model parameter choice (Fig. 6d) and model physics (Supplementary Fig. 4), repeat experiments show that the relative contributions of these variables to glacier volume change remain extremely robust (see Methods).

**Climate drivers of glacier advance and retreat**. To investigate the specific Southern Hemisphere climatic processes associated with this glacier advance phase, we analysed several climate re-analysis fields (Figs 7 and 8, Supplementary Fig. 5), regional climate indices and local synoptic-type data sets according to the upper and lower quintiles for interannual mass balance changes (Supplementary Figs 6 and 7; Supplementary Tables 4 and 5). We identified Tasman Sea surface temperature anomalies as the most consistent and significantly correlated surface climate variable throughout the mass balance year (April–March; Fig. 7). Negative anomalies in Tasman Sea surface temperature are associated with glacier mass balance gain, and positive anomalies are associated with mass balance loss.

The Tasman Sea surface temperature anomalies and glacier volume changes are associated with two large-scale features of Southern Hemisphere atmospheric circulation variability: the Pacific South American[34] and Zonal Wave 3 patterns[35] (Fig. 7). These phenomena help to enhance meridional airflow anomalies in the New Zealand sector, likely resulting in changing sea surface temperatures (Figs 7 and 8). El Niño events are on average associated with low sea surface temperatures in the New Zealand

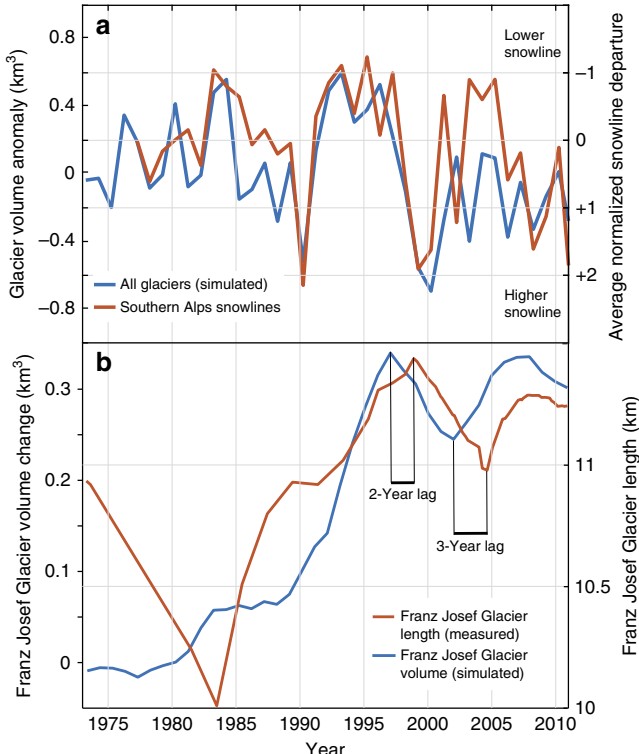

**Figure 5 | Comparison between modelled and observed glacier changes.** (**a**) Simulated interannual glacier volume anomaly (from the 'standard run') versus the observed average normalized annual snowline departure based on 26 Southern Alps glaciers (Supplementary Fig. 1). (**b**) Simulated ice volume changes for Franz Josef Glacier versus observed glacier length. Simulated ice volume changes lead to measured glacier length by 2–3 years, in accordance with observations that Franz Josef Glacier reacts swiftly to climate forcing.

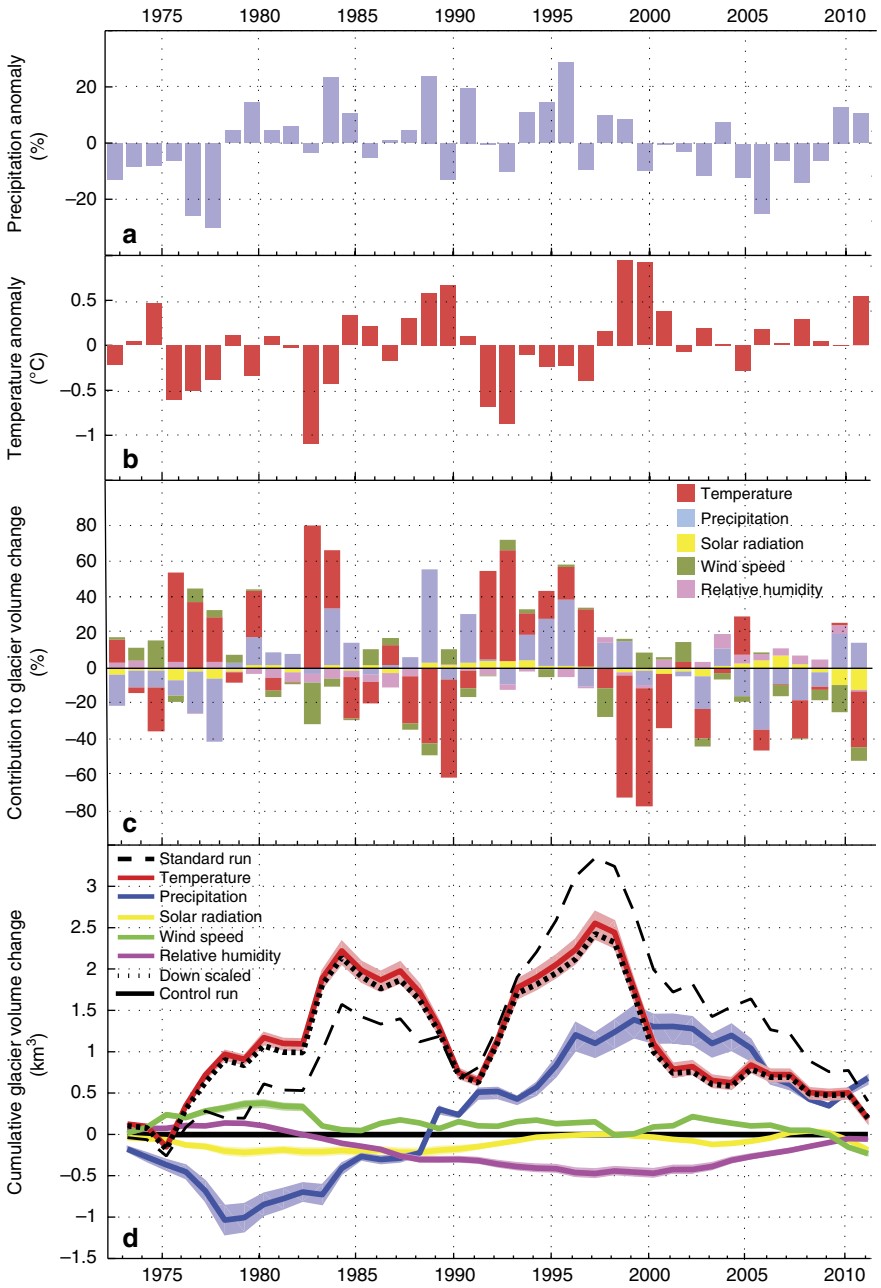

**Figure 6 | Diagnostic experiments with energy balance model.** Annual precipitation (**a**) and temperature (**b**) anomalies during the study period over the model grid. (**c**) The relative contributions of different components of the climate forcing to simulated glacier volume changes. These proportions are derived from the diagnostic experiments (**d**), which show the relative contribution of each climatic component to the cumulative glacier volume change. Overall, this figure demonstrates that reduced temperature (56%), rather than increased precipitation, made the largest contribution to glacier volume changes during this period. Precipitation contributed 27% overall, while the combined effect of wind, cloudiness and relative humidity resulted in 17% of the glacier volume change. Note that the cumulative glacier volume changes shown in **d** have been de-trended by removing the long-term ice loss signal. This is carried out by plotting each contribution relative to the control run, where all variables are held at their climatological mean values. Uncertainty bands in **d** relate to sensitivity experiments where model parameters are systematically varied (see Methods). These experiments show that the results are relatively insensitive to parameter choice. The results of a model simulation forced by downscaled temperature from climate re-analysis data[30] are also shown. This simulation falls within the uncertainty envelope of our sensitivity tests, demonstrating that the temperature forcing used in our modelling is appropriately represented by climate re-analyses[30].

region, and the period of major glacier advance in the 1980s and 1990s saw frequent El Niño activity. Yet, the oceanic and atmospheric anomalies associated with New Zealand glacier mass gain are only weakly correlated with El Niño-Southern Oscillation variability (Supplementary Fig. 6), suggesting that it is not the sole (or dominant) influence on local glacier activity or regional climate anomalies. Rather, we suggest that anomalous circulation over New Zealand required for glacial advance arises from the interplay of both tropical and extratropical processes (Supplementary Discussion), and that these circulation patterns, and associated changes in sea surface temperature (Figs 7 and 8), have broader impacts across the high southern latitudes.

The Pacific South American and Zonal Wave 3 patterns appear important for promoting glacier mass balance changes by influencing the large-scale structure of the atmospheric circulation in the New Zealand region (Figs 7 and 8, Supplementary Fig. 5 and Supplementary Discussion). During glacier mass gain years, a pattern of positive circulation anomalies associated with both the Pacific South American and Zonal Wave 3 patterns develops over the southeast Pacific, while a negative anomaly lies over New Zealand (Fig. 8). The eddy-driven jet to the south of New Zealand weakens, especially

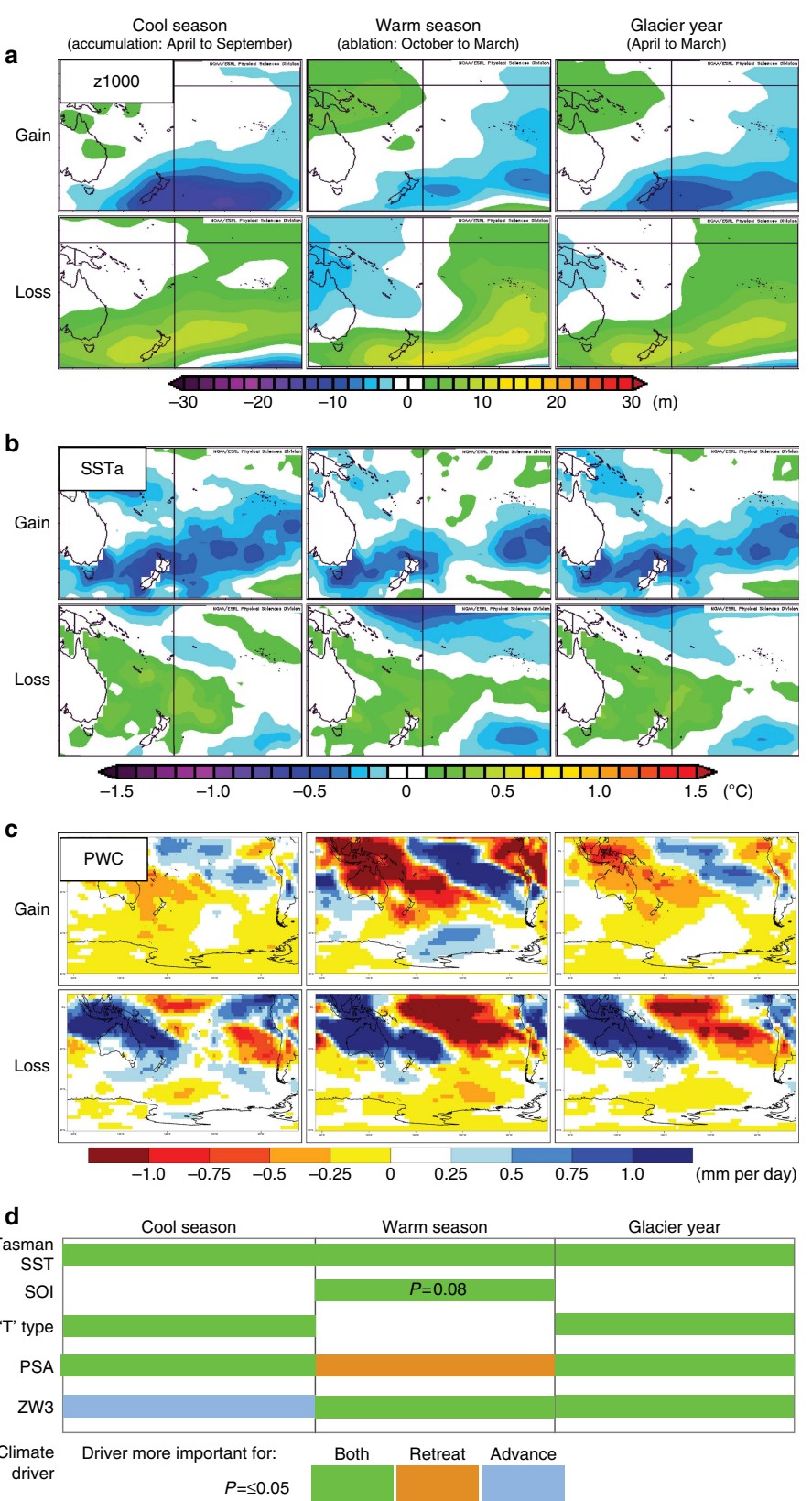

**Figure 7 | Climate anomalies for glacier mass gain and loss.** Composite patterns for upper (gain) and lower (loss) quintiles of Southern Alps glacier mass balance changes (Supplementary Table 4), for (**a**) 1,000 hPa geopotential height (analogous to atmospheric pressure near mean sea level) anomalies (z1000), (**b**) SST anomalies (SSTa), and (**c**) precipitable water content (PWC) anomalies. Glacier mass gain over cool and warm seasons, as well as the glacier year, is associated with low atmospheric height/pressure, low SST in the Tasman Sea, and slightly higher precipitation, especially in summer when it is less effective. Glacier mass loss is associated with high atmospheric height/pressure and high SST. The bottom panel (**d**) is a schematic diagram indicating significant differences in means ($P < 0.05$) for climate indices associated with glacier mass gains and losses. We assessed whether the index favoured glacier mass loss (brown), gain (blue), or both (green) by examining the differences of means (absolute values and sign, Supplementary Table 5). If the difference of means were of approximately the same magnitude and opposite sign for gain and loss, then the index (and mass balance driver) was considered important for both. If an absolute magnitude of a mean index value exceeded its counterpart by a factor of two or more, then that index (and mass balance driver) was considered more important for either gain or loss. Regional climate indices and drivers shown include Tasman SST, Southern Oscillation Index (SOI), T synoptic type ('T' type), Pacific South American pattern and Zonal Wave 3 (ZW3). Tasman SST has the strongest and most persistent association with glacier volume changes in the Southern Alps (see Supplementary Fig. 7).

during the austral cool season[36], and there is a northward displacement of the region of strongest westerlies over New Zealand during summer (Fig. 8). This configuration is associated with southerly and southwesterly wind anomalies in the Tasman Sea-New Zealand sector (Fig. 8), an increase in the frequency of deep 'lows' that pass over and south of the South Island (Supplementary Table 5)[26], and probable northward advection of cool high-latitude surface waters. The promotion of lower regional sea surface temperatures during spring and summer contributes to anomalously low temperatures in the Southern Alps (up to 1 °C lower than the 1981–2010 mean, Fig. 6b), downwind of the Tasman Sea. The lower ambient temperatures favour positive glacier mass balance by increasing the snow component of total precipitation during spring, by lowering the elevation of the temperature-dependent snow/rain threshold. Lower temperatures also reduce melt during summer, thus increasing the length of the accumulation season (Supplementary Fig. 8). Increased snow during spring also increases the glacier albedo, delaying the melt season onset and reducing melt season length (Supplementary Fig. 8).

In the last decade, a switch to more negative glacier mass balance conditions has resulted in dramatic terminus retreat of Franz Josef and Fox glaciers, especially after 2011 (ref. 8). The glacier retreat is a response to a number of negative mass balance years between 2000 and 2011, including five of the eight most negative mass balance years in our 39-year analysis. Negative glacier mass balance is caused by higher than average air temperature in our model domain. Higher temperatures are associated with anomalously high surface temperature in the Tasman Sea, and high surface pressure in the New Zealand region. These climatological conditions are promoted by the opposite polarity of the Pacific South American and Zonal Wave 3 atmospheric circulation patterns to the conditions which resulted in glacier advances (Fig. 7).

**Glacier advances and human-induced warming.** To assess whether Southern Alps glacier mass balance has been influenced by anthropogenic climate forcing during the glacier advance phase, we compare our standard run to a global-scale mass balance simulations forced by an ensemble of climate models[3] (Fig. 9). This global-scale study calculates glacier mass balance for all New Zealand glaciers, based on two sets of climate model simulations from the Coupled Model Intercomparison Project phase 5. The first set of climate model simulations includes only natural climate forcings, such as volcanic aerosols and solar variability. The second set of Coupled Model Intercomparison Project phase 5 simulations includes both natural and anthropogenic forcings, including increased greenhouse gas concentrations. We find that over the 26-year period from 1980 to 2005, which produced the climate forcing for some New Zealand glaciers to advance, the mean annual mass

balance calculated in our standard run is statistically indistinguishable from the mean annual mass balance simulated from a full range of anthropogenic and natural climate forcings[3] (Supplementary Table 6). During this same interval (1980–2005), our mean annual mass balance in the standard run is significantly different from the mean annual mass balance in the suite of mass balance simulations that only include natural climate forcings[3] (Supplementary Table 6). Further statistical tests show that this anthropogenic influence may have become more pronounced since the late 1990s (Supplementary Table 6). Together, these results suggest that Southern Alps glacier mass balance between 1980 and 2005 partly reflects anthropogenic climate change.

**Discussion**

We show that air temperature in New Zealand is the major driver of glacier mass balance during the advance phase between 1980 and 2005. We also show that Southern Alps glacier mass balance is correlated with the surface temperature of the adjacent ocean, immediately upwind of the South Island of New Zealand. A link between sea surface temperature and New Zealand glacier mass balance has previously been identified[18,37]. It has also been suggested that more frequent westerly or southerly airflow anomalies are an important driver of positive glacier mass balance in the Southern Alps[38]. Our work is consistent with previous studies, but provides a missing piece of the puzzle, by demonstrating that air temperature is the main control on glacier mass balance during this period. In contrast, previous work has focused on a range of possible climate drivers (precipitation, temperature, cloud cover and so on) without identifying the clear cause of Southern Alps mass balance changes.

Previous workers have also linked the regional airflow anomalies that drive Southern Alps glacier mass balance changes to El Niño-Southern Oscillation[38] activity. El Niño events are thought to be primarily responsible for promoting the westerly and southerly airflow anomalies that drive positive mass balance in the Southern Alps. While we agree that El Niño-Southern Oscillation plays a role in setting up the extratropical circulation patterns that ultimately affect Southern Alps glaciers, we suggest that it is only one of several contributors towards generating lower sea surface temperatures in the Tasman Sea. A strong relationship between El Niño-Southern Oscillation activity and Southern Alps glacier mass balance is not found, probably because this climate phenomenon does not always result in a predictable set of atmospheric circulation anomalies in the New Zealand region[39]. For example, during the 1997/1998 El Niño event, one of the largest of twentieth century, Southern Alps glacier mass balance was only weakly positive. This is because sea surface temperatures in the Tasman Sea, and air temperature in the Southern Alps, stayed relatively high during this event[40]. This pattern was again seen during the 2015/16 El Niño event, when the Tasman Sea and air temperature in New Zealand remained

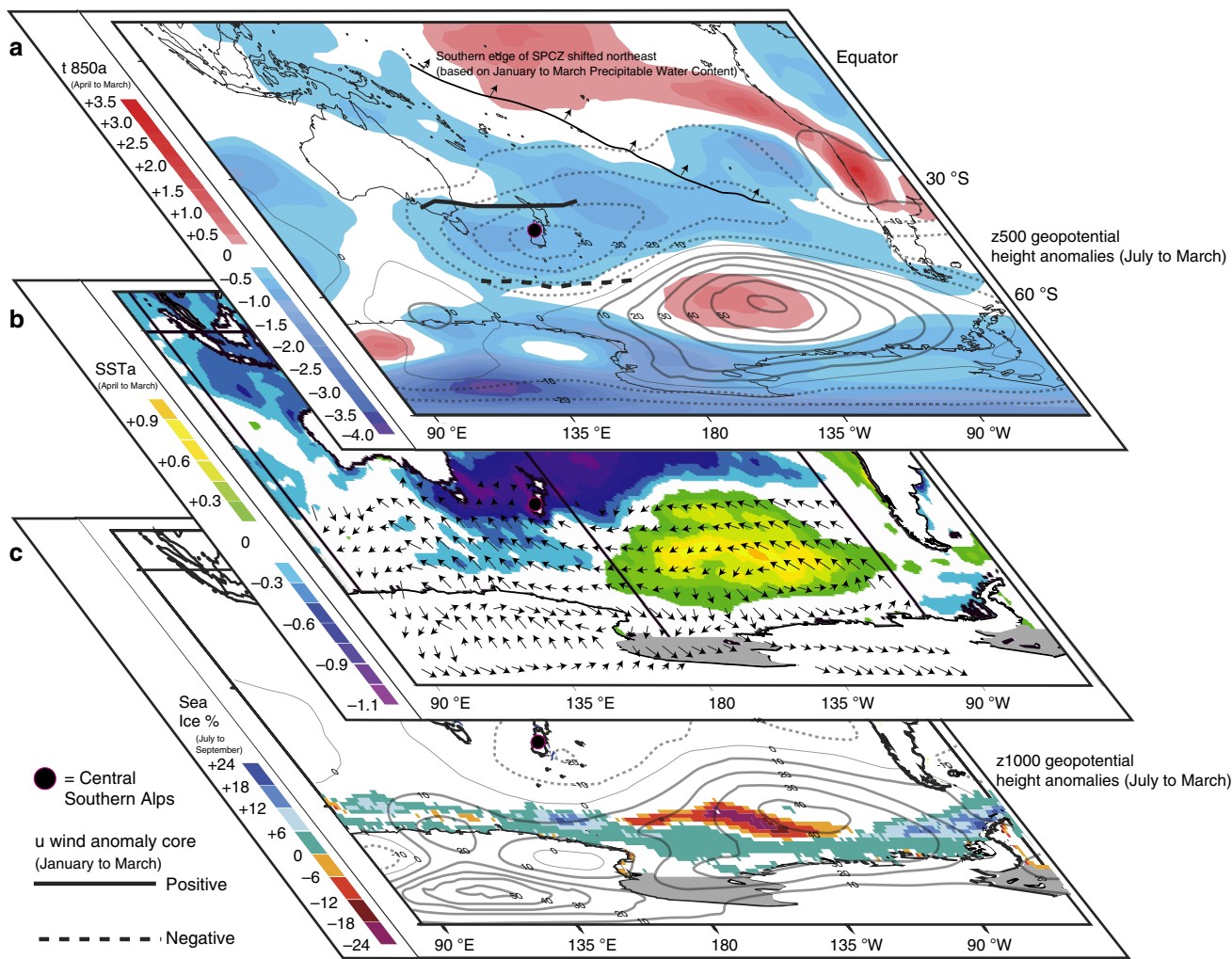

**Figure 8 | Climate anomaly pattern that drove New Zealand glacier advances.** Lower-to-middle troposphere (**a**) and surface (**b**,**c**) climate anomalies for the South Pacific region. The anomalies depict the spatial patterns for the average mass balance gain composite minus the average mass balance loss composite, and hence represent the atmospheric and oceanic conditions that promoted glacier advances in New Zealand. SPCZ refers to the South Pacific Convergence Zone. Geopotential height contours are shown every 10 m (positive = solid; negative = dashed). Zonal wind (u-wind) anomaly is at the 500 hPa geopotential height level shown in **a** and the solid (dashed) line represents the location where there is a zone of maximum (minimum) westerly anomalous flow. Arrows in **b** show surface wind anomalies. All anomalies (except sea ice) are shown relative to climatology for the 1981–2010 period. Low temperatures at 850 hPa geopotential height (t850; ~1,500 m above the sea level) extend over most of the South Pacific from southern Australia to South America. This pattern is also associated with increased sea ice concentration (%), and advancing outlet glaciers in East Antarctica directly south of New Zealand. In the Amundsen Sea sector of West Antarctica on the opposite side of this atmospheric and oceanic dipole, this climate pattern is associated with warmer air and higher SSTs and reduced sea ice concentration, and likely increases in sub-ice shelf melt rates at Pine Island Glacier.

unusually high during the austral summer[41]. Decadal-scale modulation of the El Niño Southern Oscillation cycle by the Interdecadal Pacific Oscillation is also likely to play a role, as has recently been demonstrated for Antarctic sea ice extent[42]. At present, we can only speculate about the role of the Interdecadal Pacific Oscillation because the direct relationship between El Niño Southern Oscillation activity and New Zealand glacier mass balance is weak and since our study period (39 years) is too short to obtain statistically robust results.

We find the clearest relationships between New Zealand glacier mass balance and large-scale atmospheric wave patterns in the extratropics. In particular, we identify strong associations between Southern Alps glacier mass balance and Zonal Wave 3 and the Pacific South American pattern, the latter often being associated with El Niño-Southern Oscillation activity[43,44]. Together, these climate patterns are associated with broader Southern Hemisphere impacts. For example, glacier mass gain in the Southern Alps is coincident with high pressure over

the southeast Pacific Ocean, reduced sea ice concentration in the Amundsen and Bellingshausen Seas[36,45], and increased sea ice concentration south of New Zealand (Fig. 8). We suggest that this pattern of climate variability may have also influenced Antarctic glacier mass balance, most likely via changes in wind-driven thermal ocean forcing and sub-ice shelf melting[46]. For example, in the 1990s when Southern Alps glaciers gained mass, increased westerly winds and higher ocean temperatures in the Amundsen Sea (Fig. 8) were associated with thinning of Pine Island Glacier[47], while directly south of New Zealand, stronger offshore winds from the Antarctic continent and cooling of the adjacent southwest Pacific Ocean (Fig. 8) coincided with glacier advances in coastal East Antarctica[48].

The air and sea surface temperature anomalies associated with positive glacier mass balance in the Southern Alps extend over a significant fraction of the south Pacific (Figs 7 and 8). Unsurprisingly, low temperatures are also recorded in the 'New Zealand temperature series'[49,50] during the positive glacier

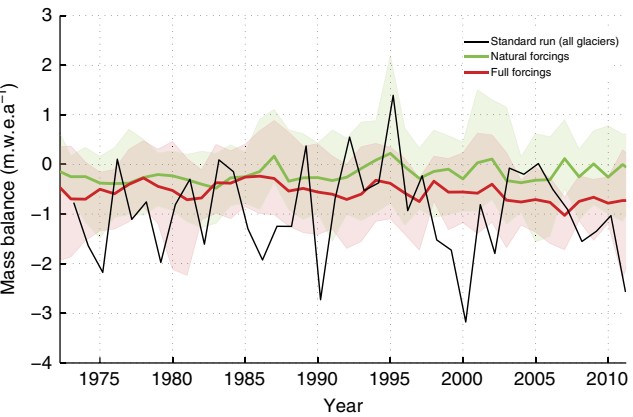

**Figure 9 | Attribution of New Zealand glacier mass balance to natural and human causes.** Standard mass balance model run (black line, all glaciers) is plotted alongside modelled New Zealand glacier mass balance output from a global-scale glacier attribution study[3]. Mass balance units are metres of water equivalent per year (m w.e.a$^{-1}$). 'Full forcings' (red) show the ensemble mean-specific mass balance and model range for New Zealand glaciers, calculated from 12 models that include both natural and anthropogenic forcings from the Coupled Model Intercomparison Project phase 5 (CMIP5). 'Natural forcings' (green) show the same for CMIP5 models that include natural and omit anthropogenic forcings on climate. The standard run derived here shows a larger degree of interannual variability than either the 'full forcing' or 'natural forcing' runs. The standard run, however, is more consistent with 'full forcing' mass balance simulation (Supplementary Table 6), suggesting that New Zealand glacier mass balance was influenced by anthropogenic climate change between 1980 and 2005.

mass balance years identified in this study. Previous work has shown that New Zealand temperature variability is strongly affected by whether the mean circulation is more northerly or more southerly[49,51]. Southerly flows have become more frequent in the New Zealand region since the 1960s, and they were especially prevalent in the early to mid-1990s when low temperatures were also recorded in the New Zealand temperature series[49]. Although we have shown that variability in mid-latitude zonal wave structures was responsible for encouraging this southerly flow, the drivers remain uncertain. Frequent El Niño events occurred in the early to mid-1990s, and we have shown that southerly airflow anomalies in the Tasman Sea region are associated with the Pacific South American Pattern, which is teleconnected to the El Niño-Southern Oscillation. However, large equatorial volcanic eruptions are also associated with low temperatures globally and with southerly airflow anomalies over New Zealand[52], and the eruption of Mt Pinatubo in 1991 may have been influential in promoting positive mass balance in the Southern Alps.

Our work builds on a global effort to attribute glacier mass balance changes to natural and anthopogenic forcings on climate[3]. In New Zealand, direct mass balance observations are lacking during the key periods (for example, the 1980s and 1990s), and model evaluation in a global attribution study[3] was carried out using data interpolated from global measurements[53] rather than from local observations. Our re-assessment confirms that New Zealand glacier mass balance was affected by anthropogenic forcing since 1980 (ref. 3). While this may seem to be a surprising result in light of the protracted advances experienced by some glaciers during this period, it is a reflection of New Zealand's location, where atmospheric and oceanic

circulation interact to produce significant interannual climate variability. Long-term trends in the New Zealand temperature series also show evidence of anthropogenic forcing, after the effects of regional circulation anomalies are removed[49].

Climate models do not fully reproduce the form or the timing of the distinct Southern Hemisphere regional atmospheric meridional airflow anomalies[49] that drive large interannual variations in New Zealand temperature, and Southern Alps glacier mass balance. Future projections of New Zealand climate, and hence glacier mass balance, are unlikely to be accurate on decadal timescales unless models capture this behaviour. The twentieth and early twenty first century behaviour of Franz Josef Glacier provides useful context for considering future predictions. Franz Josef Glacier retreated by more than 3 km during the twentieth century, in response to a warming of ~1 °C. It also experienced several discrete periods of glacier advance during this period, including the episode that we focus on in this paper (Fig. 2). Extrapolating forwards, this could mean that the future rate of ice loss could be enhanced, reduced or even temporarily reversed, depending on how regional climate drivers interact with projected centennial-scale warming. However, if anthropogenic forcing becomes dominant during the twenty first century as projected and Southern Alps temperature increases by 2 °C or more[54,55], significant Southern Alps ice loss is inevitable. The dramatic retreat of Fox and Franz Josef glaciers since 2011 may be the first sign of this rapid mass loss. Finally, we suggest that future warming of the Tasman Sea, such as that experienced in the austral summer of 2015/2016 (ref. 41), will lead to enhanced rates of ice loss, due to the close relationship between Tasman Sea surface temperature and New Zealand glacier mass balance demonstrated here.

In conclusion, we have shown that several discrete periods of reduced air temperature in the Southern Alps resulted from regional-scale climate variation over the South Pacific. This caused some New Zealand glaciers to advance between 1983 and 2008, during a period of anthropogenically forced climate warming, global-scale glacier retreat and overall ice volume loss in the Southern Alps. Our work calls into question other studies where it was concluded that precipitation increase has driven glacier advances in recent decades[6]. There is a need to carry out similar diagnostic modelling work in locations where the underlying drivers of glacier advance conditions are not well understood. Continued long-term monitoring of glacier mass balance, and high-elevation mountain climate, is also essential.

## Methods
**Glacier mass balance modelling.** The model calculates the surface mass balance of glaciers over a 38 × 40 km domain in the central Southern Alps of New Zealand (Fig. 3 and Supplementary Fig. 1), at 100 m horizontal resolution. Ablation (all processes leading to ice loss at the glacier surface, dominantly melt but also sublimation) is calculated using a spatially distributed energy balance model[10]. The model is run at a daily time step, and the results are collated for each glacier year, which starts on 1 April and ends on 31 March of the following year (end of the austral summer). The study covers a period of 39 glacier years from March 1972 to March 2011. The model is forced by daily climate data from an ~5 × 5 km (0.05° lat/lon) climate product derived from instrumental station data and climate re-analysis data.

Glacier surface mass balance depends not only on climatic variations but also on the dynamic adjustment of glacier geometry. For this reason, reference-surface balance calculated over a constant glacier geometry is better suited for climatic interpretation[56]. Thus, our glacier model considers the surface mass balance (and not the flow dynamics) of glaciers in the central part of the Southern Alps[31], (Fig. 3) over a fixed geometry. A 15 m digital elevation model for New Zealand[57] is resampled to 100 m to provide the surface elevation throughout the model domain. An ice mask and glacier debris cover are derived from aerial photography[58] captured for topographic mapping (NZ map series 260), and resampled on the same 100-m grid as the digital elevation model.

Ablation is calculated using an energy balance model[31]. Incoming short-wave radiation is derived using a diffuse and direct component[59], from calculated radiation at the top of the atmosphere[60]. A cloudiness value is inferred from

measured incoming solar radiation from the New Zealand Virtual Climate Station Network data set[28,29] to partition direct and diffuse components. The spatial distribution of short-wave radiation is calculated using topographic shading[61], and surface albedo depends on the length of time since the last snowfall[62]. Outgoing long-wave radiation ($L_{OUT}$) is calculated using the Stefan–Boltzmann law, while incoming long-wave radiation is split into an atmospheric and terrain component according to the sky view of each point on the grid. The emissivity and emission temperature of these components are different, with the atmospheric component based on air temperature, with an emissivity that incorporates assumptions about the vertical temperature profile.

Turbulent heat fluxes, $Q_H$ and $Q_C$, are calculated using a bulk aerodynamic approach, with different effective roughness length for snow and ice surfaces (Supplementary Table 1). These values were derived by tuning these effective roughness lengths until melt rates achieved a good match with 455 individual glacier mass balance measurements made on Franz Josef Glacier between 1 April 2000 and 31 March 2002. The Richardson stability criterion is used to correct the exchange coefficients for stable conditions. Precipitation heat flux, $Q_R$, is calculated assuming that precipitation is at air temperature. Subsurface heat fluxes are neglected which, while not strictly correct, is expected to have a relatively minor ($\sim 10\%$) influence on the mass balance calculation on temperate glaciers, which do not contain significant amounts of snow and ice below $0\,°C$ (ref. 59). The ground heat flux, $Q_G$, is set at $0\,W\,m^{-2}$, consistent with the assumption of surface temperature at $0\,°C$ (ref. 63). Surface debris cover is dealt with in a simple way where ablation calculated by the energy balance model is reduced by 90% where surface debris is present and when any snow accumulation on top of the debris has melted, based on observed reductions in ablation under debris on Tasman Glacier of 93% (ref. 64) and 89% (ref. 65).

To ensure that our results are not influenced by these simplifying assumptions regarding subsurface heat fluxes and melt under debris cover, the experiment is repeated with subsurface thermal calculations in snow, ice, debris and snow on debris. The evolution of temperature $T$ in the snow/ice/debris with time $t$ and at depth $z$ is calculated by solving the one-dimensional heat equation (1):

$$\rho c \frac{\partial T}{\partial t} = \frac{\partial}{\partial z}\left(k \frac{\partial T}{\partial z}\right) \tag{1}$$

where $\rho$ is the density of the material, $c$ its specific heat capacity and $k$ its conductivity (values summarized in Supplementary Table 1). The heat equation is solved independently at each grid cell using a Crank–Nicolson scheme and a standard tridiagonal solver. To resolve temperature correctly, the time step is reduced from daily (in the standard energy balance calculation) to 4 hourly to capture the diurnal evolution of the thermal state caused by changing energy fluxes at the surface. Air temperature is allowed to vary sinusoidally within the maximum and minimum temperatures of each day, while the short-wave energy components are calculated explicitly at each time step. Surface temperature is then calculated by closing the energy balance at the surface[66] using a Newton–Raphson iterative scheme[67] where the energy balance components that depend on surface temperature ($Q_H$, $Q_E$, $L_{out}$ and $Q_R$) are recalculated to estimate the derivative $dQ_s/dT_s$ of surface energy $Q_s$ with surface temperature $T_s$. The boundary conditions for the temperature calculations in each case are the same, with the surface temperature applied at the top and $0\,°C$ applied at the bottom layer. In the case of snow and ice, we are assuming that the glacier is temperate and that at 10 m below the surface there is no annual variation in temperature.

Debris thickness is poorly known in the study area, and so we take the data from Tasman Glacier[68] and extract a relationship between debris thickness and the distance from the closest clean-ice surface. This relationship is then applied to Tasman Glacier and all other debris cover in the model domain. The calculation of conductive heat flux within the debris layer generally follows ref. 67. We assume that the ice below the debris is always at melting point. This assumption is justified during the summer[67] when surface temperatures are generally above $0\,°C$. In debris cover during the winter the base of the debris will drop below $0\,°C$, but the amount of ablation under debris cover during these conditions is zero. We also assume that the base of any snow that accumulates on debris has a temperature of $0\,°C$, which also suppresses any ice ablation at the base of the debris layer. Snow on debris has no direct effect on glacier mass balance (debris cover is always in the ablation zone, so snow on debris never accumulates from year to year) but the duration of snow cover controls melt of the ice underlying the debris. Snow on debris is rather ephemeral and so this treatment is considered adequate. Ultimately, melt under debris is calculated based on the conductive heat flux at the ice/debris interface.

The data used as input to the model are minimum and maximum air temperature, solar radiation, relative humidity and precipitation. As wind speed data are not available for the whole period, wind speed at the 850-hPa level from climate re-analysis[30] is scaled to match wind speed from the New Zealand Virtual Climate Station Network for 2000–2010, and applied throughout the model period. All of the climate data fields are downscaled using bilinear interpolation from the 0.05° New Zealand Virtual Climate Station Network grid to a 100 m resolution square grid for the glacier mass balance model. Minimum and maximum temperatures are calculated from lowland stations using a seasonally variable lapse rate, which is different for minimum and maximum temperatures[28,69]. To interpolate these data to the finer glacier grid, the temperature values are first lapsed to the sea level using the same lapse rates before bilinear interpolation, and then lapsed to the 100-m grid elevations. In this way the dependence of temperature on elevation is captured, besides the regional variations in temperature.

We also carry out an experiment where the mass balance model is forced by temperature data downscaled from climate re-analysis data[30] (Fig. 6 and Supplementary Fig. 4), rather than directly from the New Zealand Virtual Climate Station Network. This allows us to assess the utility of using climate reanalysis data to identify the climate drivers of glacier advance and retreat. Downscaling is based on diagnostic regression relationships between surface temperature over New Zealand and variables taken from reanalysis data at 2.5° latitude/longitude resolution using the approach described in ref. 70. Regression relationships are developed for each standard 3-month season for surface temperature, using re-analysis 1,000 hPa height and 850 hPa temperature fields. For development of the regression equations, temperature data are taken as anomalies from daily climatological values. For maximum, minimum and mean temperature, the daily climatology accounts for up to 70% of the total variance, while the re-analysis-based regression estimates account for between 40 and 60% of the residual in western and alpine regions. Downscaled temperatures derived from re-analyses are then extrapolated to our finer-scale grid for glacier mass balance modelling, using the methods described above.

**Model evaluation and uncertainties.** To test the veracity of the climate date interpolation schemes, we compare the model input data against independently measured data at Mueller Hut and Tasman Glacier. We limit this comparison to temperature because there are very few reliable precipitation measurements in the model domain, and accumulation measurements (tested later) are the most robust way of assessing precipitation input. The Tasman Glacier temperature data[71] were collected from 2007 to 2009 at 1,139 m above the sea level, and have a root-mean square difference of 2.32 K and a bias of 0.86 K compared with our model input data. The Mueller Hut temperatures[72] were collected from 2010 to 2011 and show root-mean square difference of 2.17 K and a bias of 0.30 K. These statistics compare favourably with other temperature interpolation methods in the Southern Alps[28].

Glacier mass balance measurements made at Franz Josef and Tasman glaciers are used to tune and evaluate the glacier mass balance model. The model is also evaluated against annual glacier snowlines in the Southern Alps (a proxy for the glacier equilibrium line altitude) estimated from oblique aerial photographs since 1976 (Fig. 5), and the length change record at Franz Josef Glacier[8] (Figs 2 and 5).

Overall, 1,035 direct glacier mass balance measurements made at Franz Josef and Tasman glaciers between 1972 and 2011 are used to tune and evaluate the glacier mass balance model. In all, 455 of these mass balance measurements on Franz Josef Glacier between 1 April 2000 and 31 March 2002 are used in model tuning, while 580 measurements from before, and after, this tuning period provides an independent assessment of model performance. Overall, agreement between modelled and measured mass balance is excellent (Supplementary Fig. 2 and Supplementary Table 2). The comparison of modelled and measured accumulation also provides a secondary test of the precipitation input data.

Additional model runs were conducted to ensure the results from our diagnostic experiments were not unduly influenced by model parameter choices; the range of output values that resulted from the range of parameter choices given in Supplementary Table 1 is the blue and red uncertainty bands for precipitation and temperature, respectively, in Fig. 6d. Repeat model simulations using statistically downscaled re-analysis temperature (rather than interpolated station data) showed similar patterns (Supplementary Fig. 4) and fall within the uncertainty bounds generated using different model parameter sets (Fig. 6d).

To evaluate the influence of our simple debris-cover ablation-reduction scheme and our assumption of glacier surfaces always being at the melting point, we tested this simple scheme against our full subsurface and debris conduction calculations using the same model experiments (Supplementary Fig. 4). We found that the more complete thermal calculations gave a very similar result to our simplified analysis. Including conductive melt under debris, and subsurface heat fluxes in snow and ice, the contribution of temperature variations to volume changes is generally decreased, while the contribution of precipitation variations to volume changes is close to zero over the full period. The contribution of other variables also changes, but remains minor.

**Climate analysis.** We analysed interannual ice volume changes in the 'standard run' in order to link this signal to changes in the atmosphere and oceans. First, the control run, which contains the overall trend of volume loss over the study period, based on the mean climate, is used to de-trend the mass balance series to remove the effect of the large imbalance between present-day climate and glacier extent (shown by the generally negative mass balance in Fig. 4a). The year-to-year glacier mass balance volume anomaly variations (Fig. 5a) are ranked from largest to smallest values. The upper and lower quintiles from this ranking are then used to identify the strongest 8 years for ice volume gain and loss (Supplementary Table 4).

We calculated means and variances for several climate indices that correspond to these individual glacier years. These glacier years are further subdivided into four seasons; April, May, June (AMJ), July, August, September (JAS), October, November, December (OND) and January, February, March (JFM). The first half of the glacier year (i.e., the cool season; April to September), and the second half of the glacier year (i.e., the warm season; October to March) are also considered. Two-sample $t$-tests are applied to determine whether the means for the two groups

are significantly different at the 95% confidence level (Supplementary Figs 6 and 7). The climate indices considered in our analysis include the Southern Oscillation Index, the Niño 3.4 index, New Zealand Trenberth Indices, regional blocking indices for the southern mid-latitudes, the Southern Annular Mode, the Pacific South American mode and Zonal Wave 3 (ref. 35).

We used the Australian Bureau of Meteorology data set (http://www.bom.gov.au/climate/current/soihtm1.shtml) for the Southern Oscillation Index[73], while the Niño 3.4 (N3.4) index was sourced from a National Oceanic and Atmospheric Administration data set (http://www.cpc.ncep.noaa.gov/data/indices/ersst3b.nino.mth.ascii). The Trenberth M1 index is the normalized surface pressure difference between Hobart, Australia and Christchurch, New Zealand, and is a broad measure of meridional flow over the South Island and Tasman Sea[51]. The Trenberth MZ3 index is the normalized surface pressure difference between New Plymouth (North Island, New Zealand) and Chatham Island to the east of New Zealand and is a broad measure of South West/North East flow over the South Island of New Zealand[51]. Southern Annular Mode, Zonal Wave 3 and Pacific South American pattern index anomaly values were calculated relative to the 1981–2010 climatological normal interval using climate re-analysis data[30]. Both Zonal Wave 3 and the Pacific South American pattern are based on monthly mean 500-hPa geopotential height; the Pacific South American index is the amplitude time series of the second Empirical Orthogonal Function calculated over the southern extratropics south of 20°S, and Zonal Wave 3 is calculated according to ref. 35. The Southern Annular Mode index is calculated using normalized monthly zonal mean sea-level pressure difference between 40°S and 65°S according to ref. 74. The Southwest and Southeast Pacific Blocking Indices were calculated following ref. 75. We also analysed the monthly frequencies of occurrence of 12 New Zealand region 'Kidson' synoptic types[76] in relation to glacier variability.

Seasonal composite spatial patterns for sea surface temperature, temperature at the 850 hPa level, geopotential height anomaly at 500 and 1,000 hPa, zonal wind anomalies and precipitable water content anomalies are constructed for April to June, July to September, October to December, January to March, cool (April to September) and warm (October to March) seasons, and glacier years (April to following March) using online re-analysis from the National Oceanic and Atmospheric Administration (http://www.esrl.noaa.gov/psd/cgi-bin/data /composites/printpage.pl). All of the re-analysis data are available for the period of our mass balance simulation (1972–2011). Anomalies relative to 1981–2010 are calculated for all climate variables (Figs 7 and 8 and Supplementary Fig. 5). Sea ice concentration data extend back only to late 1981, and we utilized the National Oceanic and Atmospheric Administration monthly mean ice concentration data set available at (http://www.esrl.noaa.gov/psd/cgibin/db_search/DBSearch.Pl?) to generate spatial maps of sea ice concentration anomalies. Sea anomalies in the individual seasons in the composite plots are relative to the 1982–2010 average, and two seasons in the gain quintile (1975–1976 and 1979–1980) are therefore omitted from the compositing analysis because of temporal limitations of that data set. In addition, a sea ice concentration index for the Ross Sea sector of the Southern Ocean are utilized for cluster analysis, available from the National Snow and Ice Data Center (http://nsidc.org/data/smmr_ssmi_ancillary/area_extent.html).

**Evaluating natural and anthropogenic climate influences.** A recent global-scale attribution study of glacier mass balance changes[3] concluded that New Zealand glacier changes were consistent with mass balance model simulations forced by global climate models that included all forcings (natural and anthropogenic), rather than mass balance model simulations forced by natural forcings alone. However, a limitation of the application of this study to New Zealand glaciers[3] is the use of a global data set of glacier mass balance observations to assess their models[53]. These 'observations'[53] are 5-year averages, interpolated from global measurements during periods when no local mass balance observations are available in New Zealand (for example, the 1980s and 1990s). In this paper we provide a New Zealand-specific indication of mass balance variations for these unmeasured periods.

We compare the full ensemble and ensemble means of simulations that included all forcings, and just natural forcings, in this global attribution study[3] to those of our 'standard run' mass balance simulation to assess whether our simulations also reveal this anthropogenic influence on New Zealand glacier mass balance (Fig. 9 and Supplementary Table 6). We used two-sample $t$-tests to assess whether the means of these mass balance simulations were significantly different. We also used consecutive $t$-tests on 11-year arrays, starting from 1973 to 1983 and ending at 2001–2011 to assess whether this relationship changed through time.

**Data availability.** Additional data sets not included in the Supplementary Information are available from the corresponding author on reasonable request.

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

## Acknowledgements

This work was supported by NIWA core funding via the project 'Climate Present and Past'. We thank Ben Marzeion for providing his glacier mass balance simulations for New Zealand.

## Author contributions

A.N.M., B.M.A. and A.M.L. contributed equally; A.N.M. led the coordination and writing, B.M.A. carried out the glacier modelling and A.M.L. the climate analysis. J.A.R., P.F. and S.M.D. contributed to the climate analysis. All authors contributed to the final manuscript.

## Additional information

**Competing financial interests:** The authors declare no competing financial interests.

