## [Peer Review File · Nature Communications]

Reviewers' comments:

Reviewer #1 (Remarks to the Author):

This paper provides an interesting perspective on why glaciers were advancing in recent years in New Zealand. It applies a novel methodology using reanalyses and a glacier mass balance model with adjustments to the driving conditions of the glacier model to investigate different drivers of glacier mass balance and concludes that local temperature changes are mostly responsible for increases - including through increasing the snow component of total precipitation during spring - rather than due to increased winter precipitation. This last explanation was the one favoured in the AR5 assessment (page 338: "The exceptional terminus advances of a few individual glaciers in Scandinavia and New Zealand may be related to locally specific climatic conditions such as increased winter precipitation").

Given the interest in glacier retreat as an indicator of global change and the fact that glaciers in New Zealand have been behaving differently this paper is likely to be of widespread interest.

I think the paper is generally sound but could be communicated more clearly. I list my main concern and then some detailed comments that need addressing in revision.

Main concern

Lack of observational comparison. There are supporting comparisons of model simulations with observations in the supplementary figures but the argument in the main paper is entirely model based. This makes the overall conclusions appear generally poorly supported as the link to the real world is through some general statements early in the paper about glaciers advancing and any clear quantitative comparison between modelled and observed changes is lacking either in the figures or in the text. The key figure 2 is entirely model based so it isn't possible to judge from this how well the model does in simulating cumulative glacier volume change. Likewise Figure 3 and Figure 4 are entirely model based analyses, using reanalyses but comparing modelled glacier mass balance changes with various driving factors in the reanalyses. I suspect given the good agreement between models and observations shown in some supplementary figures that the support for deductions about what is happening in reality are better supported than they appear so that my concern does not reflect a fundamental flaw in the paper but a lack of clarity about the extent to which the modelling set up used supports the paper's conclusions. Nevertheless I think a revised paper needs to do a much better job of demonstrating the observational support for its conclusions.

Detailed comments

Line 35 It would be good to define ablation for the benefit of general readers

Line 78 This approach assumes linearity. It would be good to demonstrate clearly that linearity holds.

Lines 140-144 Figure 2 doesn't appear to show a net negative mass balance in recent years (although the mass balance is less positive than in earlier years the dashed line in Fig2 is clearly positive). Isn't this in contradiction with the statement that the mass balance has been negative between 2000 and 2011 ?

Lines 164-167 This sentence is a good example of the imprecision in the language in this paper that makes it really difficult to interpret. The focus of the paper's abstract is on advance of glaciers between 1983 and 2008 but here the text is discussing an undefined period of time over which glaciers have had negative mass balance. In fact this sentence and the following paragraph is a bit of a disaster in my view. I'm looking to the final paragraph to be a summing up of the paper in a wider context with a discussion of the main implications of the work that should add clarity for the

reader about what they have just read means. Yet this final paragraph of the paper starts with a conclusion opposite to the main conclusion of the paper and then works its way through a rather tortured logic to do with coupled climate modelling over longer timescales, differences between the approach taken in this paper and other approaches and criticisms of the inadequacies of coupled models to the final "we therefore suggest" that is the main conclusion from the paper that follows from the arguments presented earlier in the paper but not at all from this final paragraph. This last paragraph serves to confuse rather than enlighten.

In summary, the final paragraph needs complete rewriting and the rest of the text revising to ensure greater clarity including being very clear about periods of time concerned and providing comparative quantitative figures where appropriate. The arguments laid out in the main paper including the figures need to better describe the observational support for the paper's conclusions.

Reviewer #2 (Remarks to the Author):

Title: Regional cooling caused recent New Zealand glacier advances in a period of global warming
Manuscript: NCOMMS-16-08744-T

Authors: Andrew N. Mackintosh, Brian M. Anderson, Andrew M. Lorrey, James A. Renwick, Prisco Frei and Sam M. Dean

General Comment:

This manuscript proposes regional cooling was responsible for periods of glacier advance over the period 1972-2011 by comparing output from a regional-scale energy balance model for the central Southern Alps of New Zealand to large scale atmospheric circulation in the Southern Hemisphere. The motivation to do so is simple. At the end of the 20th and beginning of the 21st century a number of glaciers in the Southern Alps advanced and/or experienced positive mass balance (mass gain) during a period of global warming, which requires explanation. As noted by the authors, the last two IPCC reports have indicated that precipitation might be responsible for this advance (see further comments below). Thus, the authors target air temperature and precipitation as the explanatory variables for the observed positive balances.

The so called "debate" about whether air temperature or precipitation is responsible for glacier advance in the Southern Alps is quite old, initiated by two contrasting publications in the early 1980s (Salinger et al., 1983; air temperature, Hessel, 1983; precipitation). As noted by Chinn et al. (2005; pg. 152-153, the key reference used by IPCC to conclude that precipitation might be responsible) "correlations of the Franz Josef Glacier frontal fluctuations and climate using only temperature and precipitation were inconclusive". This brought about a shift in focus to assess the controls of atmospheric circulation on glacier mass balance, with anomalous (south) westerlies found to be responsible for higher precipitation and lower air temperatures (Fitzharris et al., 1997 and references therein), with precipitation sometimes being cited as being more dominant (e.g. Fitzharris et al., 2007, pg. 160) despite little direct evidence shown to support this conclusion. Not satisfied with the suggestions that Franz Josef Glacier is more sensitive to precipitation and changes in atmospheric circulation, Oerlemans (1997) used a numerical ice flow model to show that air temperature is likely to have the largest influence on glacier advance, which was supported in a similar study by the first two authors of the present work (Anderson and Mackintosh, 2006). Further, Anderson et al. (2010, 2012) have argued that glaciers in the Southern Alps are more sensitive to changes in air temperature using energy balance modelling. Thus, the claim that air temperature is more dominant than precipitation in controlling glacier behaviour in the Southern Alps by the lead authors of the present research is not new and has been central in a number of their publications. However, what is new is that the present research is specifically targets the recent periods of advance of some glaciers in the Southern Alps in an effort to build a case to identify the primary atmospheric and oceanic drivers.

The manuscript contains three main parts: 1. regional-scale energy and mass balance modelling, 2. climate analysis and 3. comparison to GCM-driven glacier mass balance modelling. The climate analysis identifies the importance of the Pacific South American (PSA) and Zonal Wave 3 (ZW3) patterns in controlling oceanic and atmospheric anomalies, which is interesting and new compared to previous research on the large scale atmospheric circulation controls on glacier behaviour in the Southern Alps. The linkage between this analysis and glacier mass balance is primarily statistical. Thus, for the present research to be of interest to others in the field it is critical that the authors demonstrate that the regional-scale energy balance modeling adequately resolves the key physical processes controlling mass balance, and to inform readers how air temperature and precipitation influence mass gain and loss. Thus, the focus of the following comments target this issue, which the authors may wish to consider should the paper be considered for publication in Nature Communications.

Specific comments:

Please note that page number is referred to as (P) and line number is referred to as (L).

Main paper

1. L21-24: Is there a reason why the authors have omitted reported glacier advance in Southern Patagonia (Chile)? In Vaughan et al. (2013, pg. 345, FAQ 4.2 | Are Glaciers in Mountain Regions Disappearing?) it is stated "In a few regions, however, individual glaciers are behaving differently and have advanced while most others were in retreat (e.g., on the coasts of New Zealand, Norway and Southern Patagonia (Chile), or in the Karakoram range in Asia)."

2. P2, L42-45: "Previous work has suggested a link between this glacier advance phase and atmospheric circulation changes, leading the Fourth and Fifth Intergovernmental Panel on Climate Change (IPCC) assessments to report that increased precipitation was responsible". The only publication in relation to the processes held responsible for glacier advance in the Southern Alps cited by the IPCC is Chinn et al. (2005). The references to New Zealand in the reports are:

"As with coastal Scandinavia, glaciers in the New Zealand Alps advanced during the 1990s, but have started to shrink since 2000. Increased precipitation may have caused the glacier growth (Chinn et al., 2005)" (see Lemke et al., 2007, pg. 360).

"The exceptional terminus advances of a few individual glaciers in Scandinavia and New Zealand in the 1990s may be related to locally specific climatic conditions such as increased winter precipitation (Nesje et al., 2000; Chinn et al., 2005; Lemke et al., 2007)" (see Vaughan et al., 2013, pg. 338).

"In a few regions, however, individual glaciers are behaving differently and have advanced while most others were in retreat (e.g., on the coasts of New Zealand, Norway and Southern Patagonia (Chile), or in the Karakoram range in Asia). In general, these advances are the result of special topographic and/or climate conditions (e.g., increased precipitation)." (see Vaughan et al., 2013, pg. 345, FAQ 4.2 | Are Glaciers in Mountain Regions Disappearing?).

The authors should be very clear that the Chinn et al. (2005) reference appears to have been responsible for the perception that precipitation "might be" responsible for the recent glacier advance in NZ. The IPCC reports don't explicitly state that "increased precipitation was responsible" as indicated on L44-45, and the Hooker and Fitzharris (1999) reference refers to changes in both precipitation and air temperature being responsible for glacier advance and retreat (see their conclusions). Chinn et al. (2005) had no basis to make the statement that changes in precipitation are primarily responsible for the advance of glaciers in the abstract and conclusions of their work, as they mention the importance of both air temperature and precipitation in their

discussion. "An increase in the strength of this circulation and an associated increase in precipitation together with lower air temperatures during the ablation seasons are the climatic variations responsible for the mass balance increase in both regions" (Chinn et al., 2005, pg. 154). The authors of the present manuscript should consider changing their present sentence to more carefully reflect the position of IPCC, and perhaps go as far as to mention how influential (and arguably misleading) parts of the Chinn et al. (2005) publication has been.

3. P3, L50-52: As noted by Oerlemans (2005, pg. 676), "Glacier mass balance depends mainly on air temperature, solar radiation, and precipitation. Extensive meteorological meteorological experiments on glaciers have shown that the primary source for melt energy is solar radiation but that fluctuations in the mass balance through the years are mainly due to temperature and precipitation." The authors should consider addressing the importance of solar radiation directly (not indirectly through their reference to cloudiness) and need to demonstrate more clearly its overall influence on controlling melt during summer (see below for further comments).

4. P3-5, L55-99: The authors introduce the regional-scale energy balance model, and refer readers to Supplementary Information for a full description of the model. Detailed comments about the model are provided below. The diagnostic experiments provide readers with the contribution (as percentages) of different variables to changes in glacier volume. Air temperature is identified as the dominant variable to cause glacier changes during the advance phase (56%) but the authors provide no information as to how air temperature controls mass balance and what the uncertainty of this estimate is. To make a significant contribution, some insight must be provided as to what effect air temperature has on different physical processes. For example, in what order of significance does a reduction in air temperature influence changes in albedo, melt and/or the rain/snow threshold. I don't think readers should be expected to accept the percentage contributions of each variable tested in the diagnostic experiments without insight into the modelled changes to the key physical processes governing advance or retreat. At the very least, a few key sentences describing these in the main body of the manuscript are necessary and detailed information in the Supplementary Information should be provided.

5. P5-8, L100-163: The authors identify the importance of the PSA and ZW3 patterns, which are likely controlling variability in SST - a key control on glacier mass balance. Previous research has suggested that recent glacial expansion has been controlled primarily by two inter-related climate modes. A positive phase in the Inter-decadal Pacific Oscillation (IPO) between 1978 and 1998 was thought to have had the effect of strengthening the influence of the El Niño Southern Oscillation (ENSO) in the New Zealand region, resulting in a higher frequency of El Niño events that enhanced west to south-west atmospheric circulation (Salinger et al., 2001). The authors do not mention the IPO at all, but probably should as it has been regarded as the mechanism controlling the strength and frequency of ENSO. Clarifying the relationship between IPO and the indices described in this research would be of interest to readers if the case is being made that PSA and ZW3 are the dominant climate patterns controlling SSTs and mass balance.

6. P6-7, L134-139: The authors provide some information about how lower air temperatures influence mass gain. These are very general and don't provide significant new insights into what changes occur as a result of reduced air temperatures. The authors wish to advance knowledge, but very similar statements have already been made in relation to glaciers in the Southern Alps. These comments should be much more tightly constrained (see comment 4 above) using evidence from the regional-scale atmospheric modelling - describing the relative lengths of the ablation and accumulation seasons does not reveal the key physical processes controlling mass gain and loss. Also, how is the length of an ablation season and/or accumulation season calculated - when does a season start or stop? Is it the sum of days each year that have mass gain versus loss, or is a method constructed that allows end points to be established? Please clarify as identification of the start and end of an ablation season is not that trivial.

Supplementary information

7. P3-8, L68-147: The regional-scale energy and mass balance modelling is critical in determining the relative roles of different climate variables on glacier advance and retreat and governs the key finding of the research, as described in the abstract "Here, we show that advance of glaciers in NZ between 1983 and 2008 was primarily due to reduced air temperature rather than increased precipitation". For this statement to be upheld the authors must show more evidence that the model being used is resolving the key physical processes controlling glacier behaviour, in particular the role air temperature plays in controlling mass gain and loss. The uncertainty of this estimate must also be more carefully scrutinized. To this end, the authors should consider addressing the following issues:

7.1 The model parameters used to calculate the radiation components are not described, and no validation of the cloudiness values is attempted. Their effect on model uncertainty is not addressed at all (Supplementary Table 1), which is questionable given that net radiation is likely (or should be) the largest control on ablation in summer. The role net radiation has on ablation is not stated, which it should be to provide readers assurance that the model is resolving this key component of the energy balance appropriately.

7.2 The statement that turbulent heat fluxes make up half or more of the energy available for melt in maritime environments is not correct (L90-91). This statement appears to be sourced directly from Anderson and Mackintosh (2012, Section 4.3.1). For example, values determined from energy balance modelling using automatic weather station data as input from both Norway and New Zealand clearly show that net radiation is the dominant energy source for ablation, which is governed by net shortwave radiation (e.g. Giesen et al., 2009, 2014; Cullen and Conway, 2015). Anderson et al. (2010, pg. 124) overestimated the role turbulent heat fluxes play in controlling ablation using the same model, and incorrectly stated that "radiation dominates the energy balance in winter, while turbulent fluxes dominate both in summer, when temperatures are higher, and on an annual scale". To address this problem, the authors must provide energy balance values in a table or something similar to show readers that the basic energy balance is reproduced correctly, otherwise the diagnostic experiments are likely to have an exaggerated sensitivity to air temperature.

7.3 It appears that the roughness lengths for momentum, heat and moisture are assumed to be equal, which has recently been shown not to be the case on Brewster Glacier (Conway and Cullen, 2013). Thus, the "effective" roughness length for ice (S Table 1) is an order of magnitude larger than the effective roughness length suggested by Conway and Cullen (2013), which likely leads to an overestimation of the turbulent heat fluxes. As stated by the authors, the roughness lengths were tuned until melt rates were matched with 455 individual glacier mass balance measurements. The problem with this approach is that the turbulent heat fluxes are modified until mass balance requirements are met, which comes at the expense of the more important radiation terms, which are not part of the tuning. This is likely why the relative role of the turbulent heat fluxes is suggested to be equal or greater than half of the melt energy, when in fact, net radiation on these high altitude glaciers in the central Southern Alps is very likely the largest energy source for melt. The turbulent heat fluxes at the higher elevations are unlikely to provide more than one third of the energy for melt (Cullen and Conway, 2015).

7.4 The model assumes the surface temperature is equal to melting point ($0 \text{ }^{\circ}\text{C}$), which is not appropriate in summer or any other seasonal period and can lead to uncertainties in modelled mass balance (e.g. Pelliccoitti et al., 2009, Conway and Cullen, 2013). The contribution of the subsurface heat flux should also be considered, and the assumption of it being equal to 0 W m^{-2} is not valid (Cullen and Conway, 2015).

7.5 The manner in which debris covered surfaces is dealt with is very rudimentary and not state-of-the-art. A number of models now exist that allow the surface energy balance of debris covered surfaces to be resolved (e.g. Collier et al., 2014). Anderson and Mackintosh (2012) used the same approach and acknowledged its limitations, but no effort to improve the scheme has subsequently

been attempted. The issue is addressed by including and excluding the ablation reduction scheme but these additional runs are not incorporated into an overall uncertainty (see point 3 below - diagnostic experiments and hypothesis testing).

7.6 The authors should explain how minimum and maximum air temperature are used as model input (daily) - is an average of these used to represent air temperature (P4, L188-119) or does the model cater for both a minimum and maximum air temperature. If this is the case, how are the other variables introduced into the model on a daily time scale (mean values, or something else)?

7.7 The precipitation from the VCSN product contains significant uncertainties, especially within the model domain. How well is the spatial and temporal variability of precipitation within the model domain represented? In Anderson and Mackintosh (2012) it is noted that "snow thickness is truncated at a maximum value to avoid build up of excessive snow thickness in glacier accumulation areas" (Section 4.3.3.) - is this also applied in this model set up? Are any other precipitation adjustments made to satisfy mass balance requirements? The model does not include any processes that account for the redistribution of snow or avalanching, which are known to be important for the mass balance of glaciers beneath the highest peaks in the Southern Alps. Do the VCSN interpolated precipitation data really allow you to model snowfall and mass balance without any adjustment in the highest elevation areas in the Southern Alps? If so, what are your maximum precipitation values and how do they compare to the maximum values given on P3, L63?

7.8 As noted by Anderson and Mackintosh (2012), mass balance is very sensitive to the chosen lapse rate. It is noted on P4, L127-129 "the temperature values are first lapsed to sea level using the same lapse rates before linear interpolation, and then lapsed to the 100-m grid elevations". How are the air temperatures "lapsed" to higher grid elevations after first being lapsed down to sea level? Supplementary Table 1 suggests the lapse rate is seasonally variable - is this still maintained and how? If the Norton method of interpolation is maintained, how has the documented warm bias in ablation season air temperatures been addressed (e.g. Tait and Macara, 2014)?

8. P5-6, L150-189: Model evaluation:

8.1 The model is evaluated primarily using direct mass balance measurements. Half of the measurements are used for tuning, while the remainder for validation. No input or output data are compared to automatic weather station data, which would help strengthen the validation of the atmospheric processes deemed important in controlling mass balance. If not possible, a table showing the seasonal values of the input data for the lowest and highest elevation grids (and/or the most west versus the most east grid points) over the study period would be insightful. It would certainly allow readers to ascertain how air temperature and precipitation vary, and what the seasonal range of other key meteorological variables is. Bottom line: the regional-scale atmospheric modelling as it is presented is very "black-box", and does not allow readers to get a sense of the variability of the key physical processes driving mass balance.

8.2 How are ELA departures calculated? It is not clear to this reader what is meant by "the glacier model successfully simulates both the magnitude and direction of these annual departures from the mean ELA"? A number of the glaciers in Supplementary Table 3 appear to be outside the model grid domain - how have these been used in the validation?

8.3 Could the authors clarify what "offset by approximately this amount" actually means in relation to the comparison of simulated glacier volume and measured glacier length (S Figure 3).

9. P7, L220-226: Diagnostic experiments and hypothesis testing - how are the "additional" model runs carried out and how is the assessment of total uncertainty of the model results established? Is the interaction of errors in both the parameters and input data accounted for in a meaningful way (e.g. Macguth et al., 2008)? Figure 2D suggests that uncertainty is only calculated for

individual terms, and that solar radiation, wind speed and relative humidity contain very little uncertainty. This seems very hard to believe, especially given how important solar radiation is on ablation and the uncertainty of deriving it using VCSN data products. If readers are expected to buy the suggestion that air temperature accounts for 56% of the total volume anomaly during the advance phase, the authors need to provide a more robust assessment of uncertainty that accounts for the interaction of model parameters and input data, especially as they influence some of the key physical processes controlling mass balance (e.g. air temperature effects on rain/snow threshold, ablation, albedo feedback etc.).

10. P8-9, L237-296: Climate analysis - the climate analysis is interesting and demonstrates clearly the importance of sea surface temperatures, building on the findings of Clare et al. (2002). The strength of the relationships between PSA, ZW3 and SSTa is compelling. This begs the question as to whether these regional circulation indices and sea surface temperature relationships are suitable for reconstructing air temperature and precipitation more broadly across the Southern Alps, given the sensitivity of the regional-scale energy balance model to these variables. It might be useful in the additional discussion to extend the focus beyond mass balance by describing how these findings impact our view on large scale atmospheric processes controlling weather and climate in New Zealand. How do the findings fit into our current understanding of the regional atmospheric and oceanic drivers controlling air temperature and precipitation variability in the South Island?

Minor technical suggestions

P2, L26: "These cooler air temperatures are" There is reference throughout the manuscript to warmer and/or cooler air temperatures. In the opinion of this reviewer, an air temperature can be higher or lower, but cannot be warmer or colder.

Supplementary Table 4: The variability in the mean annual input variables (minimum and maximum air temperature, solar radiation, relative humidity and precipitation) across the model domain could also be included in the table to allow readers to see how these change in the ranked (highest and lowest) mass balance years.

References

Anderson B., A. Mackintosh, 2006: Temperature change is the major driver of late-glacial and Holocene glacier fluctuations in New Zealand. *Geology*, 34(2), 121-124.

Anderson B., A. Mackintosh, D. Stumm, L. George, T. Kerr, A. Winter-Billington, S. Fitzsimons, 2010: Climate sensitivity of a high-precipitation glacier in New Zealand. *Journal of Glaciology*, 56, 114-128.

Anderson, B. and A. Mackintosh, 2012: Controls on mass balance sensitivity of maritime glaciers in the Southern Alps, New Zealand: The role of debris cover. *Journal of Geophysical Research*, 117, F01003, doi:10.1029/2011JF002064.

Chinn, T., S. Winkler, M.J. Salinger, and N. Haakensen, 2005: Recent glacier advances in Norway and New Zealand: A comparison of their glaciological and meteorological causes. *Geografiska Annaler A*, 87A, 141-157.

Clare G.R., B.B. Fitzharris, T.J.H. Chinn, M.J. Salinger, 2002: Interannual variation in end-of-summer snowlines of the Southern Alps of New Zealand, and relationships with Southern Hemisphere atmospheric and sea surface temperature patterns. *International Journal of Climatology*, 22, 107-120, doi: 10.1002/joc.722.

Collier, E., L.I. Nicholson, B.W. Brock, F. Maussion, R. Essery, A.B.G. Bush, 2014: Representing moisture fluxes and phase changes in glacier debris cover using a reservoir approach. *The*

Cryosphere, 8, 1429-1444, doi:10.5194/tc-8-1429-2014.

Conway, J.P., N.J. Cullen, 2013: Constraining turbulent heat flux parameterization over a temperate maritime glacier in New Zealand. *Annals of Glaciology*, 54(63) 41-51, doi: 10.3189/2012AoG63A604.

Cullen, N.J., J.P. Conway, 2015: A 22-month record of surface meteorology and energy balance from the ablation zone of Brewster Glacier, New Zealand. *Journal of Glaciology*, 61, 931-946.

Fitzharris, B.B., T.J.H. Chinn, G.N. Lamont, 1997: Glacier balance fluctuations and atmospheric patterns over the Southern Alps, New Zealand. *International Journal of Climatology*, 17, 1-19.

Fitzharris, B.B., G.R. Clare, J. Renwick, 2007: Teleconnections between Andean and New Zealand glaciers. *Global and Planetary Change*, 59, 159-174.

Giesen, R.H., L.M. Andreassen, J. Oerlemans, M.R. van den Broeke, 2014: Surface energy balance in the ablation zone of Langfjordjøkelen, an arctic, maritime glacier in northern Norway. *Journal of Glaciology*, 60, 57-70, doi: 10.3189/2014JoG13J063.

Giesen, R.H., L.M. Andreassen, M.R. van den Broeke, J. Oerlemans, 2009: Comparison of the meteorology and surface energy balance at Storbreen and Midtdalsbreen, two glaciers in southern Norway. *Cryosphere*, 3(1), 57-74, doi:10.5194/tc-3-57-2009.

Hessell, J.W.D. 1983: Climatic effects on the recession of the Franz Josef Glacier, New Zealand. *Journal of Science*, 26, 315-320.

Hooker, B.L., B.B. Fitzharris, 1999: The correlation between climatic parameters and the retreat and advance of Franz Josef Glacier, New Zealand. *Global and Planetary Change*, 22, 39-48.

Lemke, P., J. Ren, R.B. Alley, I. Allison, J. Carrasco, G. Flato, Y. Fujii, G. Kaser, P. Mote, R.H. Thomas and T. Zhang, 2007: Observations: Changes in Snow, Ice and Frozen Ground. In: *Climate Change 2007: The Physical Science Basis. Contribution of Working Group I to the Fourth Assessment Report of the Intergovernmental Panel on Climate Change* [Solomon, S., D. Qin, M. Manning, Z. Chen, M. Marquis, K.B. Averyt, M. Tignor and H.L. Miller (eds.)]. Cambridge University Press, Cambridge, United Kingdom and New York, NY, USA.

Machguth, H., R.S. Purves, J. Oerlemans, M. Hoelzle, F. Paul, 2008: Exploring uncertainty in glacier mass balance modelling with Monte Carlo simulation, *The Cryosphere*, 2, 191-204, doi:10.5194/tc-2-191-2008.

Nesje, A., O. Lie, and S.O. Dahl, 2000: Is the North Atlantic Oscillation reflected in Scandinavian glacier mass balance records? *Journal of Quaternary Science*, 15(6), 587-601.

Oerlemans, J., 1997: Climate sensitivity of Franz Josef Glacier, New Zealand, as revealed by numerical modelling. *Arctic and Alpine Research*, 29, 233-239.

Oerlemans, J., 2005: Extracting a climate signal from 169 glacier records. *Science*, 308, 675-677.

Pellicciotti F., M. Carenzo, J. Helbing, S. Rimkus, P. Burlando, 2009: On the role of the subsurface heat conduction in glacier energy-balance modelling. *Annals of Glaciology*, 50(50), 16-24.

Salinger, M.J., M.J. Heine, C.J. Burrows, 1983: Variations of the Stocking (Te Wae Wae) Glacier, Mount Cook, and climatic relationships. *New Zealand Journal of Science*, 26, 321-338.

Salinger, M.J., J. Renwick, A.B. Mullan, 2001: Interdecadal Pacific Oscillation and South Pacific climate. *International Journal of Climatology*, 21, 1705-1721, doi: 10.1002/joc.691.

Tait, A., G. Macara, 2014: Evaluation of interpolated daily temperature data for high elevation areas in New Zealand. *Weather and Climate*, 34, 36-49.

Vaughan, D.G., J.C. Comiso, I. Allison, J. Carrasco, G. Kaser, R. Kwok, P. Mote, T. Murray, F. Paul, J. Ren, E. Rignot, O. Solomina, K. Steffen and T. Zhang, 2013: Observations: Cryosphere. In: *Climate Change 2013: The Physical Science Basis. Contribution of Working Group I to the Fifth Assessment Report of the Intergovernmental Panel on Climate Change* [Stocker, T.F., D. Qin, G.-K. Plattner, M. Tignor, S.K. Allen, J. Boschung, A. Nauels, Y. Xia, V. Bex and P.M. Midgley (eds.)]. Cambridge University Press, Cambridge, United Kingdom and New York, NY, USA.

Reviewer #3 (Remarks to the Author):

This a highly appropriate paper for Nature Communications.

The most important reason is, in a nutshell - it is "accepted wisdom" - like a myth or belief (in my humble opinion) that glacier advances or even pauses are due only to (or mainly) precipitation over instrumental records. This has been done with little robust testing or analyses, except comparing wiggles, or comparing glacier changes (qualitative or quasi-quantitative way) with precipitation changes; however, these accepted wisdoms never bother to think that both temperature and precipitation change. It is never just one of the two. Furthermore, people then use this assumption, without rigorous testing, for implications for how people have interpreted even longer term paleo records. To me, the "it is only precip" statement is one of those assumptions that is ingrained in the literature, despite never having been thoroughly tested. I agree with Line 46 when citing the IPCC that is remains speculative.

This paper is one of the first, and robust testing of this assumption that I have seen. And, it shows when put to the test (pun intended), temperature also changed during periods when glaciers advanced, as well as precipitation. Furthermore using a model, the authors rigorously and statistically show both are responsible for glacier changes with temperature being more important.

What makes their paper stand apart is the use of a sophisticated glacier-climate model - grounded in observational testing or truthing (as they mention) - as a distinct test of the assumption, which has not been done to this extent. Will it lay to rest to all the precipitation-only people? No, but some people are stubborn and will ignore evidence they do not like. Hence, the paper will be slightly controversial, but in a good constructive way - hence also appropriate for ... Nature Communications.

The paper will also be highly relevant for societally important syntheses such as IPCC, because it can the explain glacier advances (the few and far between) punctuating net retreat over the time period of instrumental record. i.e., natural climate variability superimposed on general warming can explain the advances observed in New Zealand. One of the implications (to me) is that the IPCC got it wrong; glaciers did not advance at certain times in the 20th century because of (only) precipitation.

In terms of a Nature Communications paper, the topic will also be something the media can convey relatively easily to the broader audience.

That being said, I list minor to moderate revisions/suggestions. I think some of these will clarify

places or make the paper stronger. None of them negate anything said above.

Line 28. Given the broad audience, in the abstract, is there a way to not use a term Zonal Wave 3 (real jargon)? Or, is it absolutely necessary?

Line 34. Minor, acronyms such as NZ. This is a style issue, but is it necessary to abbreviate NZ? How many times is it used? I can see SOI, etc., but the paper is acronym heavy, which is needed, but in this case perhaps not. Up to the authors. I assume it is to keep the word count down.

Line 47-50. This could probably be clearer (two sentences?). Also, I would add something more specific to a second sentence - e.g., 'physical linkage between NZ glaciers and atmospheric characteristics and components of the....' This gets out what their contribution.

Line 55 to 60. This is one of my most important comments. I think one important aspect of their study might be confusing or non-appreciated or misinterpreted by some. That is, what they show the climate data document, versus what glaciers did and their tests on the sensitivity. That is, temperature lowered during these years - it has nothing to do with their study. Thus, I think one (or two) sentences highlighting or separating out this fact, independent of their study and glaciers. This in itself may not be appreciated in a quick read of their paper. That is, regardless of their study, it got cooler - it is not derived from their study as an inference. NZ glaciers did not just experience a precip change as, for example, the IPCC stated.

118-121. I think this sentence or how it link to the next paragraph can be stronger. Come back to SST or what controls them. More specific? Or maybe it is partly redundant with next paragraph?

130-136. This part and the associated part in the supplement also may be misinterpreted by some (?). That is, there is a precipitation issue here discussed - but it is ultimately driven by temperature. I can see some saying, but the authors contradict themselves. No, they do not. They start the sentence off by saying "the cooler ambient temperatures favor....." that is, these processes are a consequence or linked to temperature change. As said in first page, the two change together. It snows more because it is colder in spring. Less melting also, which they say. Anyway, to me they are clear (the cooler ambient temperatures favor....) but I wondered if they can be even clearer the effects are a consequence.

Line 140. Can they present the discussion in this paragraph as providing a 'test' of prior analyses and their findings/outcomes?

Line 168. Nature will have its guidelines, but can they abbreviate at least to "Supplement figure 10 and Table 6" there are no tables in main text, so it cannot be confused.

Figure 1. Need to explain what SST and sea ice mean - of what? Annual? Summer?

Figure 2. It is explained in the supplement, but they should state in the caption even briefly that the shading around the line represents...

Figure 3. I am not sure if they can do anything about this, but it is not intuitive what are "composite patterns....for 1000 hPa geopotential height anomalies," how is 1000 hPa relevant? I am just bringing this up given the very broad audience of Nature Communication. Would there be a way to better explain what this means in the caption, or text (I do not recall seeing a better explanation in the text). I am sure if a reader wades through the supplement, it can be sort of figured out, but I had a hard time, not being an expert in why the patterns relate to the 1000 hPa Geo height anomalies. For comparison, SST and PWC are more intuitive.

Also in the last sentence Tasman SST has the..... I would refer to one of the supplement figures and/or tables which show this.

Is there space for one more sentence saying other areas such as Patagonia may be the same issue for certain years (builds on their figure 4). "our study calls into question other areas where it has been concluded precip drives glaciers for some years..."

Supplement text figures/captions

This section can use some strengthening.

First, there are a lot of acronyms in the text, in the figures and in the supplement. At first I was going to insist that they need to be listed at least in one of these captions, the first time used, for example. Or in multiple captions (sup figure 9). Eventually I found them in the supplement, but these takes too long and needs to be more quickly found by Nature communication readership.

Then I thought what might be best is for them to add one more Supplement Table - with every acronym.

Line 63 - also figure 6?

69 - 71 some terms may be slightly harder to understand for non experts dynamic adjustment of glacier geometry? ...constant glacier hypsometry? Maybe add another sentence or two to explain what these things are to nonexpert.

171 - I think they mean standardized, not normalized. They are defined differently.

259. I would put it in at least one caption (or see comments below for figure 9, and/or a new table).

420 paragraph. same comment as above, maybe just add something along the lines of "however, all these are effects or are linked to a temperature decrease...."

432. Our analyses suggests or indicates...." (demonstrates seems like a strong word in the context). Up to authors.

Supp Figure 7. I think adding one sentence for the reader on the key effect/punchline would help. Explain what they are looking at in a nutshell. It is hard for the reader to take all of this in. And the comment above about Z1000 applies here also - what does it mean exactly to say it follows z1000 (?).

Supp Figure 8. What is Y axis on histograms (Only scatterplot Y axis explained)? They can Label y axis on histogram.

Acronyms defined? The reader has to wade through the text. See above comment. Maybe add a table.

Figure 9. Y axes? Add label - there is space. What are we looking at in terms of Y axis? I would also consider adding a legend on the right side. Or, can they just spell out the acronyms for this figure on the Y axis? SAM = xxxxx; ZW3 = xxxxx; etc. There is room. Or, just say in caption "acronyms in a (new) Table X." One or a few of these options would make it much easier for the reader to appreciate such figures.

Also, in general it might be questioned how much of this is their work in this paper or comparing to prior analyses? Maybe they should specifically state " We analyzed...", or "This was analyzed...."

Supplement Figure 10. I know it is in the legend, but I would consider adding 'red' afteranthropogenic components (in red). And green after forcings on climate (in green). Would be easier for reader.

Last, I do mention I am not an expert on the model, so cannot really evaluate the details of the model guts. However, it seems much of the background or context behind (at least earlier versions) the model has been published (reviewed already), so all seems fine.

Reviewer #1 (Remarks to the Author):

This paper provides an interesting perspective on why glaciers were advancing in recent years in New Zealand. It applies a novel methodology using reanalyses and a glacier mass balance model with adjustments to the driving conditions of the glacier model to investigate different drivers of glacier mass balance and concludes that local temperature changes are mostly responsible for increases - including through increasing the snow component of total precipitation during spring - rather than due to increased winter precipitation. This last explanation was the one favoured in the AR5 assessment (page 338: "The exceptional terminus advances of a few individual glaciers in Scandinavia and New Zealand may be related to locally specific climatic conditions such as increased winter precipitation").

Given the interest in glacier retreat as an indicator of global change and the fact that glaciers in New Zealand have been behaving differently this paper is likely to be of widespread interest.

I think the paper is generally sound but could be communicated more clearly. I list my main concern and then some detailed comments that need addressing in revision.

Main concern

Lack of observational comparison. There are supporting comparisons of model simulations with observations in the supplementary figures but the argument in the main paper is entirely model based. This makes the overall conclusions appear generally poorly supported as the link to the real world is through some general statements early in the paper about glaciers advancing and any clear quantitative comparison between modelled and observed changes is lacking either in the figures or in the text. The key figure 2 is entirely model based so it isn't possible to judge from this how well the model does in simulating cumulative glacier volume change. Likewise Figure 3 and Figure 4 are entirely model based analyses, using reanalyses but comparing modelled glacier mass balance changes with various driving factors in the reanalyses. I suspect given the good agreement between models and observations shown in some supplementary figures that the support for deductions about what is happening in reality are better supported than they appear so that my concern does not reflect a fundamental flaw in the paper but a lack of clarity about the extent to which the modelling set up used supports the paper's conclusions. Nevertheless I think a revised paper needs to do a much better job of demonstrating the observational support for its conclusions.

We thank Reviewer 1 for generally supporting our paper. Our paper was originally formatted for Nature Climate Change, and the more generous content allowance of Nature Communications has allowed us to do a more thorough job of explaining the observational support for our conclusions. Specifically;

- *We clearly define the 'glacier advance phase' in the introduction of the paper, by using glacier length changes at Franz Josef Glacier as a benchmark.*

- *The new introduction includes a fuller description of the contrasting responses of different glacier types in the Southern Alps. This also serves to clarify why some glaciers advanced between 1983 and 2008, while many others retreated.*
- *We include a new section in the 'Results' named 'Simulated and observed glacier mass balance between 1972 and 2011' which describes our standard mass balance model run, and how it compares to observations.*
- *We include two new figures (Fig. 2 and Fig. 4) that shows measured glacier length changes, as well as modelled glacier volume changes for a number of New Zealand glaciers.*
- *We bring our model evaluation figure from the former supplementary text in the main manuscript, to show that the model does an excellent job of simulating year-to-year variation in glacier mass balance, as well as the volume changes that led to changes in the length of Franz Josef Glacier.*
- *Our former Figure 1 has been split into two figures, and the model domain (Fig. 3) now shows the spatial distribution of mean modelled glacier mass balance, to demonstrate that our modelling captures the strong gradients in glacier mass balance within the Mt Cook region.*

Detailed comments

Line 35 It would be good to define ablation for the benefit of general readers

Ablation is now defined in the Methodology section (line 440) and 'melt' is used in the main part of the paper, where we describe situations which are dominated by melt rather than other ablation terms such as sublimation.

Line 78 This approach assumes linearity. It would be good to demonstrate clearly that linearity holds.

Our analysis does indeed rely on the assumption that each of the individual forcing's effect on glacier volumes are linearly additive. We confirm in the text that for the period of this analysis the individual forcing effects sum to the volume changes driven by the standard run. We also demonstrate in Supplementary Figure 3 that there is no correlation between temperature and precipitation, such that they can be treated as independent. While neither of these provide comprehensive proof of linear additivity such an assumption appears reasonable within the constraints of this experiment. Further testing could be achieved by undertaking a large suite of additional experiments involving longer simulations, larger forcings etc. However, our understanding of the physics of the glacier system suggest these are unlikely to change the assumption in any meaningful way. We note that, in the field of detection and attribution of climate change, widely relied upon by the IPCC, the same assumption is used almost universally for different anthropogenic forcing experiments. Many studies have tested this assumption in detail, and found it to be robustly true for temperature, but less so for other modelled variables such as precipitation. Such deviations from strict additivity have not prevented the methodology from being useful or widely applied. As such, while we agree with the reviewer, we see little real

benefit in doing additional analysis at this stage.

Lines 140-144 Figure 2 doesn't appear to show a net negative mass balance in recent years (although the mass balance is less positive than in earlier years the dashed line in Fig2 is clearly positive). Isn't this in contradiction with the statement that the mass balance has been negative between 2000 and 2011?

We agree that this was confusing. Our new introduction and 'Simulated and observed glacier mass balance between 1972 and 2011' sections explain this apparent contradiction. Our new Figure 4 shows how some glaciers in our model domain gain mass during this period (e.g Franz Josef, Fox), while others (Tasman, Murchison) lose volume. Figure 4 also shows that the raw glacier mass balance for all glaciers in our domain has indeed been largely negative between 2000-2011. Note, however, that we do not say that 'glacier mass balance has been negative between 2000 and 2011'. We say 'The glacier retreat is a response to a number of negative mass balance years between 2000 and 2011, including five of the eight most negative mass balance years in our 39-year analysis.' These negative years were mostly at the start and the end of the decade, and mass balance was briefly positive in the middle of the decade (which is why Franz Josef Glacier continued to advance until 2008).

The glacier mass balance variations in (former Figure 2, now Fig. 6) are detrended. We previously stated this in the caption 'The long-term ice loss signal in Panel D is removed by plotting each contribution relative to the control run, where all variables are held at their climatological mean values.' We now clarify this further by saying 'Note that the cumulative glacier volume changes shown in (D) have been de-trended, by removing the long-term ice loss signal. This is carried out by plotting each contribution relative to the control run, where all variables are held at their climatological mean values.'

Lines 164-167 This sentence is a good example of the imprecision in the language in this paper that makes it really difficult to interpret. The focus of the paper's abstract is on advance of glaciers between 1983 and 2008 but here the text is discussing an undefined period of time over which glaciers have had negative mass balance. In fact this sentence and the following paragraph is a bit of a disaster in my view. I'm looking to the final paragraph to be a summing up of the paper in a wider context with a discussion of the main implications of the work that should add clarity for the reader about what they have just read means. Yet this final paragraph of the paper starts with a conclusion opposite to the main conclusion of the paper and then works its way through a rather tortured logic to do with coupled climate modelling over longer timescales, differences between the approach taken in this paper and other approaches and criticisms of the inadequacies of coupled models to the final "we therefore suggest" that is the main conclusion from the paper that follows from the arguments presented earlier in the paper but not at all from this final paragraph. This last paragraph serves to confuse rather than enlighten.

We apologise for this and expect that part of the problem came from the condensed format.

We kept these comments in mind at all times while rewriting the introduction and when writing the Discussion section. Specifically;

- *We now define the 'glacier advance phase' at the start of the paper, and refer to years and periods in all cases where it may be ambiguous.*
- *The mass balance attribution experiment now has its own section in the results.*
- *The overall picture that some glaciers gained mass, while the majority lost mass is now clearer.*
- *The Discussion section has a different paragraph dedicated to each of the main results of this study, with a focus on how we have advanced knowledge.*

In summary, the final paragraph needs complete rewriting and the rest of the text revising to ensure greater clarity including being very clear about periods of time concerned and providing comparative quantitative figures where appropriate. The arguments laid out in the main paper including the figures need to better describe the observational support for the paper's conclusions.

We have paid attention to all of these suggestions and we hope that Reviewer 1 finds the new version of our paper easier to follow and more grounded in observations.

Reviewer #2 (Remarks to the Author):

Title: Regional cooling caused recent New Zealand glacier advances in a period of global warming

Manuscript: NCOMMS-16-08744-T

Authors: Andrew N. Mackintosh, Brian M. Anderson, Andrew M. Lorrey, James A. Renwick, Prisco Frei and Sam M. Dean

General Comment:

This manuscript proposes regional cooling was responsible for periods of glacier advance over the period 1972-2011 by comparing output from a regional-scale energy balance model for the central Southern Alps of New Zealand to large scale atmospheric circulation in the Southern Hemisphere. The motivation to do so is simple. At the end of the 20th and beginning of the 21st century a number of glaciers in the Southern Alps advanced and/or experienced positive mass balance (mass gain) during a period of global warming, which requires explanation. As noted by the authors, the last two IPCC reports have indicated that precipitation might be responsible for this advance (see further comments below). Thus, the authors target air temperature and precipitation as the explanatory variables for the observed positive balances.

The so called "debate" about whether air temperature or precipitation is responsible for glacier advance in the Southern Alps is quite old, initiated by two contrasting publications in the early 1980s (Salinger et al., 1983; air temperature, Hessel, 1983; precipitation). As noted by Chinn et al. (2005; pg. 152-153, the key reference used by IPCC to conclude that precipitation might be responsible) "correlations of the Franz Josef Glacier frontal fluctuations and climate using only

temperature and precipitation were inconclusive". This brought about a shift in focus to assess the controls of atmospheric circulation on glacier mass balance, with anomalous (south) westerlies found to be responsible for higher precipitation and lower air temperatures (Fitzharris et al., 1997 and references therein), with precipitation sometimes being cited as being more dominant (e.g. Fitzharris et al., 2007, pg. 160) despite little direct evidence shown to support this conclusion. Not satisfied with the suggestions that Franz Josef Glacier is more sensitive to precipitation and changes in atmospheric circulation, Oerlemans (1997) used a numerical ice flow model to show that air temperature is likely to have the largest influence on glacier advance, which was supported in a similar study by the first two authors of the present work (Anderson and Mackintosh, 2006). Further, Anderson et al. (2010, 2012) have argued that glaciers in the Southern Alps are more sensitive to changes in air temperature using energy balance modelling. Thus, the claim that air temperature is more dominant than precipitation in controlling glacier behaviour in the Southern Alps by the lead authors of the present research is not new and has been central in a number of their publications. However, what is new is that the present research is specifically targets the recent periods of advance of some glaciers in the Southern Alps in an effort to build a case to identify the primary atmospheric and oceanic drivers.

The extra space afforded by Nature Communications allows us to more adequately acknowledge this previous literature (in the Introduction), and articulate how we have advanced knowledge (in the Discussion section).

The manuscript contains three main parts: 1. regional-scale energy and mass balance modelling, 2. climate analysis and 3. comparison to GCM-driven glacier mass balance modelling. The climate analysis identifies the importance of the Pacific South American (PSA) and Zonal Wave 3 (ZW3) patterns in controlling oceanic and atmospheric anomalies, which is interesting and new compared to previous research on the large scale atmospheric circulation controls on glacier behaviour in the Southern Alps. The linkage between this analysis and glacier mass balance is primarily statistical. Thus, for the present research to be of interest to others in the field it is critical that the authors demonstrate that the regional-scale energy balance modeling adequately resolves the key physical processes controlling mass balance, and to inform readers how air temperature and precipitation influence mass gain and loss. Thus, the focus of the following comments target this issue, which the authors may wish to consider should the paper be considered for publication in Nature Communications.

We have responded to all detailed comments (below) in order to demonstrate that the regional energy balance modelling adequately resolves the key physical processes controlling mass balance.

Specific comments:

Please note that page number is referred to as (P) and line number is referred to as (L).

Main paper

1. L21-24: Is there a reason why the authors have omitted reported glacier advance in Southern Patagonia (Chile)? In Vaughan et al. (2013, pg. 345, FAQ 4.2 | Are Glaciers in Mountain Regions Disappearing?) it is stated "In a few regions, however, individual glaciers are behaving differently and have advanced while most others were in retreat (e.g., on the coasts of New Zealand, Norway and Southern Patagonia (Chile), or in the Karakoram range in Asia)."

We have added this reference to Patagonia. There was no reason for omitting it other than because it is less well documented than in other places. There is an opportunity to apply our approach to south Patagonian glaciers, and we point this out in the final paragraph of the paper (following advice from Reviewer 3).

2. P2, L42-45: "Previous work has suggested a link between this glacier advance phase and atmospheric circulation changes, leading the Fourth and Fifth Intergovernmental Panel on Climate Change (IPCC) assessments to report that increased precipitation was responsible". The only publication in relation to the processes held responsible for glacier advance in the Southern Alps cited by the IPCC is Chinn et al. (2005). The references to New Zealand in the reports are:

"As with coastal Scandinavia, glaciers in the New Zealand Alps advanced during the 1990s, but have started to shrink since 2000. Increased precipitation may have caused the glacier growth (Chinn et al., 2005)" (see Lemke et al., 2007, pg. 360).

"The exceptional terminus advances of a few individual glaciers in Scandinavia and New Zealand in the 1990s may be related to locally specific climatic conditions such as increased winter precipitation (Nesje et al., 2000; Chinn et al., 2005; Lemke et al., 2007)" (see Vaughan et al., 2013, pg. 338).

"In a few regions, however, individual glaciers are behaving differently and have advanced while most others were in retreat (e.g., on the coasts of New Zealand, Norway and Southern Patagonia (Chile), or in the Karakoram range in Asia). In general, these advances are the result of special topographic and/or climate conditions (e.g., increased precipitation)." (see Vaughan et al., 2013, pg. 345, FAQ 4.2 | Are Glaciers in Mountain Regions Disappearing?).

The authors should be very clear that the Chinn et al. (2005) reference appears to have been responsible for the perception that precipitation "might be" responsible for the recent glacier advance in NZ. The IPCC reports don't explicitly state that "increased precipitation was responsible" as indicated on L44-45, and the Hooker and Fitzharris (1999) reference refers to changes in both precipitation and air temperature being responsible for glacier advance and retreat (see their conclusions). Chinn et al. (2005) had no basis to make the statement that changes in precipitation are primarily responsible for the advance of glaciers in the abstract and conclusions of their work, as they mention the importance of both air temperature and precipitation in their discussion. "An increase in the strength of this circulation and an

associated increase in precipitation together with lower air temperatures during the ablation seasons are the climatic variations responsible for the mass balance increase in both regions" (Chinn et al., 2005, pg. 154). The authors of the present manuscript should consider changing their present sentence to more carefully reflect the position of IPCC, and perhaps go as far as to mention how influential (and arguably misleading) parts of the Chinn et al. (2005) publication has been.

We now cite the paper that clearly led to the IPCC summation (Chinn et al. 2005) and also cite Fitzharris et al (2007) for the reason stated by the reviewer above. We also directly quote from the IPCC report to avoid ambiguity about what the IPCC did or did not say.

3. P3, L50-52: As noted by Oerlemans (2005, pg. 676), "Glacier mass balance depends mainly on air temperature, solar radiation, and precipitation. Extensive meteorological meteorological experiments on glaciers have shown that the primary source for melt energy is solar radiation but that fluctuations in the mass balance through the years are mainly due to temperature and precipitation." The authors should consider addressing the importance of solar radiation directly (not indirectly through their reference to cloudiness) and need to demonstrate more clearly its overall influence on controlling melt during summer (see below for further comments).

We agree that the text in Lines 50-52 was somewhat misleading, underplaying the role of solar radiation. We've replaced the opening sentence with a paraphrasing of the text from Oerlemans above (acknowledged).

In the modelling, the importance of solar radiation is addressed directly. Cloudiness controls how solar radiation is split into direct and diffuse components, and the incoming longwave contribution, so is a necessary part of the calculation.

To address the request to be more upfront about how radiation affects mass balance in the model, we have added the relative contributions of different components of the energy balance to mass balance to the text (line 135).

4. P3-5, L55-99: The authors introduce the regional-scale energy balance model, and refer readers to Supplementary Information for a full description of the model. Detailed comments about the model are provided below. The diagnostic experiments provide readers with the contribution (as percentages) of different variables to changes in glacier volume. Air temperature is identified as the dominant variable to cause glacier changes during the advance phase (56%) but the authors provide no information as to how air temperature controls mass balance and what the uncertainty of this estimate is. To make a significant contribution, some insight must be provided as to what effect air temperature has on different physical processes. For example, in what order of significance does a reduction in air temperature influence changes in albedo, melt and/or the rain/snow threshold.

I don't think readers should be expected to accept the percentage contributions of each variable tested in the diagnostic experiments without insight into the modelled changes to the key physical processes governing advance or retreat. At the very least, a few key sentences describing these in the main body of the manuscript are necessary and detailed information in the Supplementary Information should be provided.

We have added a new sentence in the main part of the manuscript 'While these exact percentage contributions of different climate drivers vary slightly depending on model parameter choice (Fig. 6D) and model physics (Supplementary Figure 8), repeat experiments shows that the relative contributions of these variables to glacier volume change remain extremely robust (see methodology section).' See more on this below.

The influence of air temperature on different physical processes which affect mass balance is well understood, and this study does not have any unique insights to provide in that regard. The processes and how they are influenced by changes in air temperature have been quite well traversed in this maritime environment by Anderson et al. (2010) and Conway and Cullen (2016).

5. P5-8, L100-163: The authors identify the importance of the PSA and ZW3 patterns, which are likely controlling variability in SST - a key control on glacier mass balance. Previous research has suggested that recent glacial expansion has been controlled primarily by two inter-related climate modes. A positive phase in the Inter-decadal Pacific Oscillation (IPO) between 1978 and 1998 was thought to have had the effect of strengthening the influence of the El Niño Southern Oscillation (ENSO) in the New Zealand region, resulting in a higher frequency of El Niño events that enhanced west to south-west atmospheric circulation (Salinger et al., 2001). The authors do not mention the IPO at all, but probably should as it has been regarded as the mechanism controlling the strength and frequency of ENSO. Clarifying the relationship between IPO and the indices described in this research would be of interest to readers if the case is being made that PSA and ZW3 are the dominant climate patterns controlling SSTs and mass balance.

Previous workers have pointed out a relationship between the phase of the Interdecadal Pacific Oscillation, and New Zealand glacier advance and retreat. As the reviewer points out, the Interdecadal Pacific Oscillation is believed to represent low-frequency modulation of the El Niño Southern Oscillation. In particular, positive phases of the Interdecadal Pacific Oscillation are typically associated with more frequent El Niño events, and hence, positive glacier mass balance. While we agree that the positive mass balance years that caused glacier advances in the Southern Alps mostly fall within a single (positive) phase of the Interdecadal Pacific Oscillation, we can presently only speculate about whether this climate oscillation is an important control on glacier mass balance. This is for three reasons (1) the direct relationship between the El Niño Southern Oscillation and New Zealand glacier mass balance is weak. (2) We are unable to examine this relationship statistically because our study period (39 years) is of similar length to this oscillation (~20-40 years) and (3) The physics of the Interdecadal Pacific Oscillation are not well understood. We added a brief explanation of these issues to the text (line 350).

6. P6-7, L134-139: The authors provide some information about how lower air temperatures influence mass gain. These are very general and don't provide significant new insights into what changes occur as a result of reduced air temperatures. The authors wish to advance knowledge, but very similar statements have already been made in relation to glaciers in the Southern Alps. These comments should be much more tightly constrained (see comment 4 above) using evidence from the regional-scale atmospheric modelling - describing the relative lengths of the ablation and accumulation seasons does not reveal the key physical processes controlling mass gain and loss. Also, how is the length of an ablation season and/or accumulation season calculated - when does a season start or stop? Is it the sum of days each year that have mass gain versus loss, or is a method constructed that allows end points to be established? Please clarify as identification of the start and end of an ablation season is not that trivial.

As discussed under point 4, we consider that the processes that result in glacier mass gain when temperatures are lower is well established and is it not the aim of this manuscript to go over that ground again. We have summarised this understanding more clearly at line 275 of the revised manuscript:

'The lower ambient temperatures favour positive glacier mass balance by increasing the snow component of total precipitation during spring, by lowering the elevation of the temperature-dependent snow/rain threshold. Lower temperatures also reduce melt during summer, thus increasing the length of the accumulation season (Supplementary Figure 7). Increased snow during spring also increases the glacier albedo, delaying the melt season onset and reducing melt season length (Supplementary Figure 7).'

The way that the lengths of the ablation and accumulation seasons are calculation is now described in the caption for Figure S7: "As season length varies with elevation, we selected a site near the long-term equilibrium line of Tasman Glacier (1740 m above sea level) to examine changes in the length of the accumulation and ablation seasons."

Supplementary information

7. P3-8, L68-147: The regional-scale energy and mass balance modelling is critical in determining the relative roles of different climate variables on glacier advance and retreat and governs the key finding of the research, as described in the abstract "Here, we show that advance of glaciers in NZ between 1983 and 2008 was primarily due to reduced air temperature rather than increased precipitation". For this statement to be upheld the authors must show more evidence that the model being used is resolving the key physical processes controlling glacier behaviour, in particular the role air temperature plays in controlling mass gain and loss. The uncertainty of this estimate must also be more carefully scrutinized. To this end, the authors should consider addressing the following issues:

7.1 The model parameters used to calculate the radiation components are not described, and no validation of the cloudiness values is attempted. Their effect on model uncertainty is not

addressed at all (Supplementary Table 1), which is questionable given that net radiation is likely (or should be) the largest control on ablation in summer. The role net radiation has on ablation is not stated, which it should be to provide readers assurance that the model is resolving this key component of the energy balance appropriately.

The radiation calculation comprises four components – incoming longwave, outgoing longwave, incoming shortwave and outgoing shortwave. These calculations have been described in detail in Anderson et al. (2010) and Anderson and Mackintosh (2012), referred to in the original text. The parameterisations used are simple, but well established in the literature, as described in those papers. Cloudiness directly influences incoming long-wave and shortwave radiation.

Incoming longwave radiation depends on cloudiness, and we use the parameterisation of Konzelmann et al. (1994) to estimate incoming longwave radiation. Recent work (Conway et al., 2015) has shown that this parameterisation is appropriate.

Outgoing long-wave radiation depends primarily on surface temperature. To address the reviewer’s concerns here and later about our initial assumption that snow and ice surfaces are always at 0 °C we have implemented a surface temperature calculation scheme (described fully in the Methods section). On average this reduces outgoing longwave radiation because radiation is proportional to the fourth power of the surface temperature.

Incoming short-wave radiation depends on cloudiness, shading, and time of day. Each of these components are calculated explicitly by the model.

Outgoing short-wave radiation depends primarily on the albedo of the glacier surface. The scheme used for albedo calculation here (Oerlemans and Knap, 1998) is appropriate and captures the evolution of albedo after snow fall and through the ablation season. There are uncertainties in the parameters used in this scheme. Conway and Cullen (2016) used the same scheme at Brewster Glacier (not within our study area) but with different (higher) albedo values for the three different surface types (snow, firn, ice). Without any albedo data within our study area to test the parameterisations properly, and given that the standard values from Oerlemans and Knap (1998) provided an adequate fit for our earlier study at Brewster Glacier, we use these values here. The lower values of albedo in our study increase the importance of solar radiation.

We acknowledge that there are uncertainties in each of these four components of the net radiation. By implementing a surface temperature scheme we have addressed the largest uncertainty in the outgoing longwave radiation. The outgoing shortwave radiation is controlled using a scheme and parameters that have been tested on Brewster Glacier (Anderson et al., 2010) and found to work acceptably well.

The reviewer’s concern seems to be mainly with the treatment of cloudiness which influences the incoming short and longwave radiation. There are two aspects to this treatment – first, the dataset which provides solar radiation data, and second the way in which this is used to infer cloudiness.

We use an interpolated data product (VCSN) which provides a daily total energy from solar radiation. The interpolation is done from station data using a trivariate spline, where the third variable (after x and y) is a map of mean annual cloudiness made from MODIS imagery. The maximum error from this dataset for 'median annual daily solar irradiance' at the stations is $0.83 \text{ MJ m}^{-2} \text{ day}^{-1}$ (Tait and Zhang, 2007). However, larger uncertainties arise from the interpolation over a sparsely-measured mountainous area. The approach of using a mean annual cloudiness map captures some of this spatial variability. We acknowledge that this is not a perfect dataset. However, it does capture the broad-scale temporal and spatial variability of cloudiness and is the best dataset available. Further work improving this dataset would of course be valuable.

We infer cloudiness from this dataset using the method described by Hock (2005), and applied at Brewster Glacier by Anderson et al. (2010) – that is, developing a relationship between cloudiness and the fraction of top of atmosphere radiation to measured (or interpolated) radiation on the ground. This relationship is then used at each timestep to estimate cloudiness which then partitions the incoming shortwave radiation into direct and diffuse components, and provides the basis for the incoming longwave radiation calculation.

In response to the reviewer's concerns, we have made the following changes

- 1. the overall approach for calculating net radiation is now explained in more detail in the Methods section;*
- 2. a surface temperature scheme has been implemented to address issues with the outgoing longwave radiation;*
- 3. as well as the existing sensitivity tests for albedo (which controls the outgoing shortwave radiation) we have added further sensitivity tests for other aspects of the albedo calculation, and for cloudiness directly (which influences incoming longwave and shortwave radiation); and*
- 4. we have added the contribution of overall energy balance components to the text so that the reader can assess the influence of difference components.*

7.2 The statement that turbulent heat fluxes make up half or more of the energy available for melt in maritime environments is not correct (L90-91). This statement appears to be sourced directly from Anderson and Mackintosh (2012, Section 4.3.1). For example, values determined from energy balance modelling using automatic weather station data as input from both Norway and New Zealand clearly show that net radiation is the dominant energy source for ablation, which is governed by net shortwave radiation (e.g. Giesen et al., 2009, 2014; Cullen and Conway, 2015). Anderson et al. (2010, pg. 124) overestimated the role turbulent heat fluxes play in controlling ablation using the same model, and incorrectly stated that "radiation dominates the energy balance in winter, while turbulent fluxes dominate both in summer, when temperatures are higher, and on an annual scale". To address this problem, the authors must provide energy balance values in a table or something similar to show readers that the basic energy balance is reproduced correctly, otherwise the diagnostic experiments are likely to have an exaggerated sensitivity to air temperature.

In response to these reviewer's comments we have:

- 1. changed the wording to make it clear that turbulent fluxes do not always dominate the energy balance, even in maritime environments;*
- 2. changed 'roughness length' to 'effective roughness length' in the manuscript*
- 3. added a table which shows the relative importance of different energy balance components to melt*
- 4. as well as the existing sensitivity tests for albedo (which controls the outgoing shortwave radiation) we have added further sensitivity tests for other aspects of the albedo calculation, and for cloudiness directly (which influences incoming longwave and shortwave radiation).*

Further detail is provided below.

There is a long line of literature which supports the conclusions that turbulent heat fluxes are an important, or even dominant, part of the energy balance in maritime climates (e.g. Hock, 2005 Table 2). Of course there is large degree of variability in the balance between different energy sources which depends on the particular site and time period, even within a relatively small area, and this explains part of the differences seen in previous studies on NZ glaciers and snowpacks. The work which has been done at Brewster Glacier demonstrates this variability. Anderson et al. (2010) using a distributed energy balance model found a slight (52%) dominance of turbulent heat fluxes. Gillett and Cullen (2011), using data from an AWS on the glacier, found a slight (52%) dominance of net radiation. However, more recent work (Cullen and Conway, 2015) showed that, again at an AWS site, net radiation dominated the energy balance during periods of melt.

The comments and conclusions written in a different paper (Anderson et al., 2010) are not the subject of this review and are only relevant inasmuch as they might point to issues in the current manuscript. Providing the relative proportions of energy for melt has been provided by many studies in the past (e.g. as tabulated in Hock, 2005) and we have now added these values to the text. However, the values cannot be directly compared to any study of energy balance at a point as the values are averaged over all of the glacier surface in the domain.

To address the reviewer's comments, we have:

- 1. Removed the text 'Turbulent fluxes....which may make up half or more of the energy available for melt in maritime environments'.*
- 2. Provided figures for the energy balance components in the text.*

7.3 It appears that the roughness lengths for momentum, heat and moisture are assumed to be equal, which has recently been shown not to be the case on Brewster Glacier (Conway and Cullen, 2013). Thus, the "effective" roughness length for ice (S Table 1) is an order of magnitude larger than the effective roughness length suggested by Conway and Cullen (2013), which likely leads to an overestimation of the turbulent heat fluxes. As stated by the authors, the roughness lengths were tuned until melt rates were matched with 455 individual glacier mass balance measurements. The problem with this approach is that the turbulent heat fluxes are modified until mass balance requirements are met, which comes at the expense of the more important

radiation terms, which are not part of the tuning. This is likely why the relative role of the turbulent heat fluxes is suggested to be equal or greater than half of the melt energy, when in fact, net radiation on these high altitude glaciers in the central Southern Alps is very likely the largest energy source for melt. The turbulent heat fluxes at the higher elevations are unlikely to provide more than one third of the energy for melt (Cullen and Conway, 2015).

It has long been understood that the roughness lengths for momentum, heat and moisture are not, in general, equal and that the roughness length for momentum are smaller than those of heat and moisture by one or two orders of magnitude (Hock, 2005). Braithwaite (1995) introduced the term 'effective roughness length' where each of the roughness values are assumed to be the same, so that the exchange coefficients have the same values as if they were calculated using the separate roughness values. This is the approach taken by many, if not most, energy balance modelling studies as there are precious few measurements of roughness length, and it is not clear how these would be spatially distributed.

The difficulties in estimating the effective roughness are why we considered this to be the primary uncertainty in the energy balance and target it by tuning the effective roughness to match ablation measurements. There is some equifinality here – different parameter sets (including radiation parameters) could result in a similar match. We could have taken the approach suggested by the reviewer in prescribing the effective roughness length and adjusting some radiation parameters instead. We addressed this issue in Anderson and Mackintosh (2012) using the same mass balance measurement dataset to test model output against. Anderson and Mackintosh (2012) tested five parameters which control albedo as it is arguably the most important control on net radiation. The parameters which control the timescale of albedo decay (d_c and t_c) had little impact on goodness of fit, or mass balance output. Variations in the basic albedo values could be used to obtain as good a match to measurement, but only by increasing the fresh snow albedo to 1.0, which is unphysical.

To describe these glaciers as 'high elevation glaciers' is a questionable generalisation given that the elevation range in the model domain goes from the very lowest glacial ice in NZ (Franz Josef Glacier terminus; 310 m a.s.l.) to the very highest (summit of Aoraki Mt Cook, 3722 m a.s.l. on the 100-m resampled grid). The energy available for melt at high elevation is rather low and does not dominate the overall energy balance. At low elevations, for example on the tongue of Franz Josef Glacier, the energy available for melt is dominated by energy from rainfall (Q_R), and the turbulent heat fluxes (Q_H and Q_E) where and when it has been measured (e.g. Marcus et al., 1985).

This discussion clearly highlights the uncertainties in energy balance calculations for the largest glaciers in the Southern Alps, and indeed for many poorly-measured glacierised parts of the world. There is clearly a need for much more detailed, and long term studies, such as those carried out recently at Brewster Glacier (e.g. Conway and Cullen, 2013; Cullen and Conway, 2015) at sites throughout our study area to refine our understanding. We acknowledge that these uncertainties may mean that our energy balance calculations may not precisely simulate reality. However, our overall results, the estimates of the contribution of various climatic input

variables, are robust under a large number of different energy balance parameter scenarios.

7.4 The model assumes the surface temperature is equal to melting point (0 {degree sign}C), which is not appropriate in summer or any other seasonal period and can lead to uncertainties in modelled mass balance (e.g. Pelliccoitti et al., 2009, Conway and Cullen, 2013). The contribution of the subsurface heat flux should also be considered, and the assumption of it being equal to 0 W m⁻² is not valid (Cullen and Conway, 2015).

7.5 The manner in which debris covered surfaces is dealt with is very rudimentary and not state-of-the-art. A number of models now exist that allow the surface energy balance of debris covered surfaces to be resolved (e.g. Collier et al., 2014). Anderson and Mackintosh (2012) used the same approach and acknowledged its limitations, but no effort to improve the scheme has subsequently been attempted. The issue is addressed by including and excluding the ablation reduction scheme but these additional runs are not incorporated into an overall uncertainty (see point 3 below - diagnostic experiments and hypothesis testing).

We acknowledge that subsurface heat flux can be an important control especially on the timing of melt of snow and ice surfaces. We also accept that our debris cover scheme is rather simple, although we consider that we had demonstrated that our overall conclusions were sound, notwithstanding the limitations in this scheme.

To address these concerns, we have implemented a full sub-surface thermal scheme that works for snow, ice, debris and snow on debris surfaces. This adds a substantial amount of computational complexity to the model. Even though we have parallelised the sub-surface thermal calculations they still take an order of magnitude longer than the original daily calculation, mainly because the thermal calculation is undertaken at a four-hourly timestep, and iteration is required to calculate surface temperature which the surface-temperature dependent energy balance components to be recalculated multiple times. Most of the other energy balance components, including solar radiation and shading have to be also recalculated at this higher resolution. Hence it is not feasible to carry out the full range of sensitivity tests within the scope of this revision. Each run involves a 40-year integration of the model for six scenarios (one control run, and five runs, one for each variable). Each sensitivity test (16 in total) requires the full 40-year, six scenario calculation. Consequently, we have presented the full thermal calculation as a sensitivity test (in Figure S8), in much the same way that the debris scenarios were presented in the original manuscript.

In response to the reviewer's comments we have:

- 1. implemented a full sub-surface thermal calculation which iterates to solve for the surface temperature,*
- 2. added a detailed explanation of the scheme to the Methods section*
- 3. added a sensitivity test which shows the results of the full sub-surface calculation.*

7.6 The authors should explain how minimum and maximum air temperature are used as model input (daily) - is an average of these used to represent air temperature (P4, L188-119) or does the model cater for both a minimum and maximum air temperature. If this is the case, how are

the other variables introduced into the model on a daily time scale (mean values, or something else)?

Air temperature is taken as a single value for each day, as a mean of the minimum and maximum temperature. Each of the other variables is taken as the VCSN value for that day. For the full thermal calculation air temperature is allowed to vary in a sinusoidal manner between the minimum and maximum daily air temperature for the sub-daily time steps.

In response to the reviewer's comments we have included a fuller description of the input data time step.

7.7 The precipitation from the VCSN product contains significant uncertainties, especially within the model domain. How well is the spatial and temporal variability of precipitation within the model domain represented? In Anderson and Mackintosh (2012) it is noted that "snow thickness is truncated at a maximum value to avoid build up of excessive snow thickness in glacier accumulation areas" (Section 4.3.3.) - is this also applied in this model set up? Are any other precipitation adjustments made to satisfy mass balance requirements? The model does not include any processes that account for the redistribution of snow or avalanching, which are known to be important for the mass balance of glaciers beneath the highest peaks in the Southern Alps. Do the VCSN interpolated precipitation data really allow you to model snowfall and mass balance without any adjustment in the highest elevation areas in the Southern Alps? If so, what are your maximum precipitation values and how do they compare to the maximum values given on P3, L63?

We acknowledge that, in common with many mountain ranges of the world, the Southern Alps have a sparse network of climate stations and that there are significant uncertainties in some of the climate input data. As there are very few reliable measurements of precipitation at high elevation there are not sufficient data to test modelled input precipitation against measured precipitation. We consider the most reliable measure of precipitation, or at least the effective precipitation which adds mass to glaciers, is by comparing modelled snow accumulation against measured values. While, again, there are limited snow accumulation measurements, those made by Anderson et al (2006) on Franz Josef Glacier and Chinn on Tasman Glacier are consistent with the precipitation values used from VCSN (Figure S2).

To answer the other questions of the reviewer

(a) yes, snow thickness is truncated, otherwise over long runs the snow thickness will increase to >200 m which is not realistic. Note that this has no direct effect on mass balance – mass balance is not 'truncated'. The effect on mass balance occurs when there is a heavy melt year and the previous winter's snow is completely melted, which cannot occur if unrealistically deep snowpacks are allowed to build up;

(b) no precipitation adjustments are made;

(c) avalanching and wind redistribution of snow are important processes which can have a significant impact on small glaciers. We do not model them here and do not consider that these processes would make more than a minor difference to our model output, but would incur

significant additional computational resources.

(d) no precipitation values are given at P3, L63. Modelled accumulation and ablation values are consistent with those given in the text.

7.8 As noted by Anderson and Mackintosh (2012), mass balance is very sensitive to the chosen lapse rate. It is noted on P4, L127-129 "the temperature values are first lapsed to sea level using the same lapse rates before linear interpolation, and then lapsed to the 100-m grid elevations". How are the air temperatures "lapsed" to higher grid elevations after first being lapsed down to sea level? Supplementary Table 1 suggests the lapse rate is seasonally variable - is this still maintained and how? If the Norton method of interpolation is maintained, how has the documented warm bias in ablation season air temperatures been addressed (e.g. Tait and Macara, 2014)?

We use the seasonally-variable lapse rates of Norton (1985) as implemented in VCSN. Tait and Macara (2014) assessed VCSN against one independent, high elevation site (near the terminus of Brewster Glacier, outside of our model domain). A more recent attempt at interpolating lowland temperatures to high elevations also included the Brewster Glacier terminus site as an independent test over a different time period (Jobst et al., 2016). The comparison between observed and interpolated data is shown in Table R1.

We have compared the input data that the energy balance model uses (i.e. after all processing and downscaling from VCSN) against that measured at two high elevation sites within our model domain (Mueller Hut, -43.72154 S, 170.06493 E, 1818 m a.s.l.; Tasman Glacier 1376469E 5171398N 1139 m a.s.l.). The Muller Hut site has only been running since 2010, which only gives an overlap of one year with our model run (we now report these numbers in the text). However, in response to this reviewer request, we have extended the comparison to 2010-2015. The Tasman Glacier site ran from March 2007 until March 2009. The results of this comparison are shown in the table and Figures below. Gaps in the table are where those data are not provided in the published papers.

Table R1.

	n	T _{min} mean difference	T _{min} RMSE	T _{max} mean difference	T _{min} RMSE	T _{mean} mean difference	T _{min} RMSE	
Brewster Glacier terminus (2004-2009)		1.03	2.79	1.52	3.07			Tait and Macara (2014)
Brewster Glacier terminus (2010-2013)	1021						2.27	Jobst et al. (2016)

Tasman Glacier (2007-2009)	608					0.86	2.32	This study
Mueller Hut (2010-2011)	315					0.30	2.17	This study
Mueller Hut (2010-2015)	1326					0.86	2.81	This study

The comparison between our input temperature data, which is based on the VCSN Norton product, shows that a warm bias does exist in the VCSN data that is $<1^{\circ}\text{C}$ at two sites within our model domain. The RMSE between our temperature input compares favourably with a recent attempt at interpolating data to high elevations near Brewster Glacier. The VCSN dataset seems to perform significantly better in our model domain than it does at the previously-documented Brewster Glacier terminus site.

The warm bias in our temperature interpolations may mean that simulated mass balance is more negative than reality. However, we note the good match between our simulated and measured mass balance (Figure S2). Further, even if our simulated mass balance is slightly too negative, this does not change the results from our anomaly analysis (Figure 6D) because the anomalies are the differences between two model runs which means that systematic differences are removed.

Figure R1. Temperature measured at Mueller Hut, and the temperature interpolated from lowland data using the methods described in the text.

Figure R2. Temperature measured at Tasman Glacier, and the temperature interpolated from lowland data using the methods described in the text.

8. P5-6, L150-189: Model evaluation:

8.1 The model is evaluated primarily using direct mass balance measurements. Half of the measurements are used for tuning, while the remainder for validation. No input or output data are compared to automatic weather station data, which would help strengthen the validation of the atmospheric processes deemed important in controlling mass balance. If not possible, a table showing the seasonal values of the input data for the lowest and highest elevation grids (and/or the most west versus the most east grid points) over the study period would be insightful. It would certainly allow readers to ascertain how air temperature and precipitation vary, and what the seasonal range of other key meteorological variables is. Bottom line: the regional-scale atmospheric modelling as it is presented is very "black-box", and does not allow readers to get a sense of the variability of the key physical processes driving mass balance.

We have now shown the comparison between temperature input data to the model and independent measurements from two sites at Tasman Glacier and Mueller Hut. The statistics describing this comparison are in the main text, as is the way in which each of the climatic input

datasets have been derived.

8.2 How are ELA departures calculated? It is not clear to this reader what is meant by "the glacier model successfully simulates both the magnitude and direction of these annual departures from the mean ELA"? A number of the glaciers in Supplementary Table 3 appear to be outside the model grid domain - how have these been used in the validation?

The reviewer is correct that we have used glaciers from outside of the model domain to create the time series of ELA (snowline) departures against which the model is compared. This is justified based on the very strong correlation between ELA departures at different glaciers (Chinn et al, 2005; Table 4), and the small number of index glaciers within the model domain not all of which can be measured each year. Hence, the wider glacier sample gives a better record of the temporal variations in snowline.

We have removed the 'magnitude and direction' sentence in the caption for Figure 5.

8.3 Could the authors clarify what "offset by approximately this amount" actually means in relation to the comparison of simulated glacier volume and measured glacier length (S Figure 3).

This comment was simply intended to show that there is a time lag between glacier volume changes simulated by the model and the length changes recorded at Franz Josef Glacier, and that this time lag is approximately the same as the 3-4 year 'reaction time' estimated by Purdie et al. 2014. We have clarified this sentence and have now marked the time lag on the figure (now Figure 5).

9. P7, L220-226: Diagnostic experiments and hypothesis testing - how are the "additional" model runs carried out and how is the assessment of total uncertainty of the model results established? Is the interaction of errors in both the parameters and input data accounted for in a meaningful way (e.g. Macguth et al., 2008)? Figure 2D suggests that uncertainty is only calculated for individual terms, and that solar radiation, wind speed and relative humidity contain very little uncertainty. This seems very hard to believe, especially given how important solar radiation is on ablation and the uncertainty of deriving it using VCSN data products. If readers are expected to buy the suggestion that air temperature accounts for 56% of the total volume anomaly during the advance phase, the authors need to provide a more robust assessment of uncertainty that accounts for the interaction of model parameters and input data, especially as they influence some of the key physical processes controlling mass balance (e.g. air temperature effects on rain/snow threshold, ablation, albedo feedback etc.).

The additional runs are carried out by running the entire modelling experiment with each parameter changed as in supplementary Table 1. This approach gives a good idea of the uncertainty in the overall results that is implied by the uncertainties in the model parameters. While these parameters could be combined in different ways to come up with wider uncertainty

bands the probability of extreme parameter values combining to give significantly different results becomes increasingly small. While it would be ideal to test parameter and input data uncertainty together in a more complete way, which would give a probability distribution function to allow a fuller assessment of uncertainty, this is not practical because of the multiple model runs required for each parameter test. The 56% that air temperature variations contribute to glacier volume changes over the period clearly has an uncertainty associated with it.

The parameter sensitivity tests as presented do include the key physical processes and feedbacks that the reviewer mentions.

The uncertainty bands in Figure 6D (was Figure 2D) are not the uncertainty in each of the climate input variables. Figure 6D shows the relative contributions of the variations in each climate parameter to the overall mass balance through the 39-year period. The relatively small contribution from relative humidity and solar radiation is because the year-to-year variability in these variability is rather small (e.g. Table S4) . The narrow uncertainty bands are because the parameters for which model sensitivity is tested do not change these contributions very much.

To address the reviewers concerns we have added more parameters which control net radiation to the sensitivity tests, including cloudiness.

10. P8-9, L237-296: Climate analysis - the climate analysis is interesting and demonstrates clearly the importance of sea surface temperatures, building on the findings of Clare et al. (2002). The strength of the relationships between PSA, ZW3 and SSTa is compelling. This begs the question as to whether these regional circulation indices and sea surface temperature relationships are suitable for reconstructing air temperature and precipitation more broadly across the Southern Alps, given the sensitivity of the regional-scale energy balance model to these variables. It might be useful in the additional discussion to extend the focus beyond mass balance by describing how these findings impact our view on large scale atmospheric processes controlling weather and climate in New Zealand. How do the findings fit into our current understanding of the regional atmospheric and oceanic drivers controlling air temperature and precipitation variability in the South Island?

We have now clarified that (also following advice from Reviewer 3) that the temperature changes that caused glacier advances in New Zealand affected a wide region. We do not, however, feel that it is within the scope of this paper to discuss the large-scale atmospheric processes controlling weather and climate in New Zealand.

Minor technical suggestions

P2, L26: "These cooler air temperatures are" There is reference throughout the manuscript to warmer and/or cooler air temperatures. In the opinion of this reviewer, an air temperature can be higher or lower, but cannot be warmer or colder.

We have changed warmer/cooler to higher/lower throughout the manuscript.

Supplementary Table 4: The variability in the mean annual input variables (minimum and maximum air temperature, solar radiation, relative humidity and precipitation) across the model domain could also be included in the table to allow readers to see how these change in the ranked (highest and lowest) mass balance years.

Thank you for this suggestion. These values have been added to Table S4.

Reviewer #3 (Remarks to the Author):

This a highly appropriate paper for Nature Communications.

The most important reason is, in a nutshell - it is "accepted wisdom" - like a myth or belief (in my humble opinion) that glacier advances or even pauses are due only to (or mainly) precipitation over instrumental records. This has been done with little robust testing or analyses, except comparing wiggles, or comparing glacier changes (qualitative or quasi-quantitative way) with precipitation changes; however, these accepted wisdoms never bother to think that both temperature and precipitation change. It is never just one of the two. Furthermore, people then use this assumption, without rigorous testing, for implications for how people have interpreted even longer term paleo records. To me, the "it is only precip" statement is one of those assumptions that is ingrained in the literature, despite never having been thoroughly tested. I agree with Line 46 when citing the IPCC that it remains speculative.

This paper is one of the first, and robust testing of this assumption that I have seen. And, it shows when put to the test (pun intended), temperature also changed during periods when glaciers advanced, as well as precipitation. Furthermore using a model, the authors rigorously and statistically show both are responsible for glacier changes with temperature being more important.

What makes their paper stand apart is the use of a sophisticated glacier-climate model - grounded in observational testing or truthing (as they mention) - as a distinct test of the assumption, which has not been done to this extent. Will it lay to rest to all the precipitation-only people? No, but some people are stubborn and will ignore evidence they do not like. Hence, the paper will be slightly controversial, but in a good constructive way - hence also appropriate for ... Nature Communications.

The paper will also be highly relevant for societally important syntheses such as IPCC, because it can explain glacier advances (the few and far between) punctuating net retreat over the time period of instrumental record. i.e., natural climate variability superimposed on general warming can explain the advances observed in New Zealand. One of the implications (to me) is that the IPCC got it wrong; glaciers did not advance at certain times in the 20th century because of (only) precipitation.

In terms of a Nature Communications paper, the topic will also be something the media can convey relatively easily to the broader audience.

We thank the reviewer for their positive comments. We respond to their suggestions below.

That being said, I list minor to moderate revisions/suggestions. I think some of these will clarify places or make the paper stronger. None of them negate anything said above.

Line 28. Given the broad audience, in the abstract, is there a way to not use a term Zonal Wave 3 (real jargon)? Or, is it absolutely necessary?

The abstract has been rewritten and generalised following Nature Communications style, and we no longer refer to Zonal Wave 3 in the abstract.

Line 34. Minor, acronyms such as NZ. This is a style issue, but is it necessary to abbreviate NZ? How many times is it used? I can see SOI, etc., but the paper is acronym heavy, which is needed, but in this case perhaps not. Up to the authors. I assume it is to keep the word count down.

We thank the reviewer for this suggestion. We have removed nearly all of the acronyms in the paper. The more generous length allowance of Nature Communications makes this possible.

Line 47-50. This could probably be clearer (two sentences?). Also, I would add something more specific to a second sentence - e.g., 'physical linkage between NZ glaciers and atmospheric characteristics and components of the....' This gets out what their contribution.

We have split this into two sentences. We have also added a significantly longer introduction which explains the background to the scientific problem in more detail, including a quote from the last IPCC report. We hope that the reviewer now finds this section to be clearer.

Line 55 to 60. This is one of my most important comments. I think one important aspect of their study might be confusing or non-appreciated or misinterpreted by some. That is, what they show the climate data document, versus what glaciers did and their tests on the sensitivity. That is, temperature lowered during these years - it has nothing to do with their study. Thus, I think one (or two) sentences highlighting or separating out this fact, independent of their study and glaciers. This in itself may not be appreciated in a quick read of their paper. That is, regardless of their study, it got cooler - it is not derived from their study as an inference. NZ glaciers did not just experience a precip change as, for example, the IPCC stated.

We thank the reviewer for pointing out this opportunity. We have added a new paragraph (line 371 onwards) to address this issue. Additionally, we have pointed out that the air and sea surface temperature anomalies depicted in (now) Figure 8 extend over a large part of the South Pacific.

118-121. I think this sentence or how it link to the next paragraph can be stronger. Come back

to SST or what controls them. More specific? Or maybe it is partly redundant with next paragraph?

These sentences lead the reader into the next paragraph but to make them more specific, we have added a reference to Figures 7 and 8 here.

130-136. This part and the associated part in the supplement also may be misinterpreted by some (?). That is, there is a precipitation issue here discussed - but it is ultimately driven by temperature. I can see some saying, but the authors contradict themselves. No, they do not. They start the sentence off by saying "the cooler ambient temperatures favor...." that is, these processes are a consequence or linked to temperature change. As said in first page, the two change together. It snows more because it is colder in spring. Less melting also, which they say. Anyway, to me they are clear (the cooler ambient temperatures favor....) but I wondered if they can be even clearer the effects are a consequence.

To clarify this further, we have split the critical sentence in two and added 'by lowering the elevation of the temperature-dependent snow/rain threshold' when describing the temperature effect on precipitation.

Line 140. Can they present the discussion in this paragraph as providing a 'test' of prior analyses and their findings/outcomes?

We would prefer not to use the discussion of these negative mass balance years as a test of our climate hypotheses because our analysis requires all 39 in order to carry out appropriate statistical tests. However, on lines 346 and 421, we now mention the 2016 El Nino event when Southern Alps glaciers experienced negative mass balance due to anomalously warm sea surface temperatures in the Tasman Sea. One year is not long enough to be a true 'test', but the behavior of glaciers in 2016 is certainly consistent with our findings.

Line 168. Nature will have its guidelines, but can they abbreviate at least to "Supplement figure 10 and Table 6" there are no tables in main text, so it cannot be confused.

Figure 1. Need to explain what SST and sea ice mean - of what? Annual? Summer?

We have added this information to the caption.

Figure 2. It is explained in the supplement, but they should state in the caption even briefly that the shading around the line represents...

We have added a description of these bands at the end of the revised figure caption.

Figure 3. I am not sure if they can do anything about this, but it is not intuitive what are "composite patterns....for 1000 hPa geopotential height anomalies," how is 1000 hPa relevant? I am just bringing this up given the very broad audience of Nature Communication.

Would there be a way to better explain what this means in the caption, or text (I do not recall seeing a better explanation in the text). I am sure if a reader wades through the supplement, it can be sort of figured out, but I had a hard time, not being an expert in why the patterns relate to the 1000 hPa Geo height anomalies. For comparison, SST and PWC are more intuitive.

1000 hPa geopotential height anomalies show variability in surface pressure at or close to sea level. In Figure (now) 7 (upper panel), blue shows low pressure while green shows high pressure. We have now included 'near sea level' in brackets before 'geopotential height anomalies (z1000)'. To make this absolutely clear, we have also described the pressure, temperature and precipitation patterns associated with glacier mass gain and loss.

Also in the last sentence Tasman SST has the..... I would refer to one of the supplement figures and/or tables which show this.

We now refer the reader to Supplementary Figure 6 at the end of this caption.

Is there space for one more sentence saying other areas such as Patagonia may be the same issue for certain years (builds on their figure 4). "our study calls into question other areas where it has been concluded precip drives glaciers for some years...

Thank you for this suggestion. The final paragraph of the paper now concludes with such a statement.

Supplement text figures/captions

This section can use some strengthening.

First, there are a lot of acronyms in the text, in the figures and in the supplement. At first I was going to insist that they need to be listed at least in one of these captions, the first time used, for example. Or in multiple captions (sup figure 9). Eventually I found them in the supplement, but these takes too long and needs to be more quickly found by Nature communication readership.

We have removed all acronyms in the main text and methodology sections, and have removed most acronyms from the supplement.

Then I thought what might be best is for them to add one more Supplement Table - with every acronym.

Line 63 - also figure 6?

We now refer to this figure in the appropriate location.

69 - 71 some terms may be slightly harder to understand for non experts dynamic adjustment of

glacier geometry? ...constant glacier hypsometry? Maybe add another sentence or two to explain what these things are to nonexpert.

We have simplified this language.

171 - I think they mean standardized, not normalized. They are defined differently.

The snowline departures are normalised. See Chinn et al. 2012 (ref 48 in main text for details).

259. I would put it in at least one caption (or see comments below for figure 9, and/or a new table).

Done (and acronyms have generally been removed)

420 paragraph. same comment as above, maybe just add something along the lines of "however, all these are effects or are linked to a temperature decrease...."

This relationship has been clarified in the main text by discussing the temperature control on the snow/rain threshold.

432. Our analyses suggests or indicates...." (demonstrates seems like a strong word in the context). Up to authors.

Changed to 'suggests.'

Supp Figure 7. I think adding one sentence for the reader on the key effect/punchline would help. Explain what they are looking at in a nutshell. It is hard for the reader to take all of this in. And the comment above about Z1000 applies here also - what does it mean exactly to say it follows z1000 (?).

We have added 'This figure shows the seasonal components of the atmospheric and oceanographic conditions that promote glacier mass gain and mass loss in the Southern Alps.'

Supp Figure 8. What is Y axis on histograms (Only scatterplot Y axis explained)? They can Label y axis on histogram.

Explanations have now been provided in the Figure captions.

Acronyms defined? The reader has to wade through the text. See above comment. Maybe add a table.

They have been removed or clearly defined on first usage.

Figure 9. Y axes? Add label - there is space. What are we looking at in terms of Y axis? I would

also consider adding a legend on the right side. Or, can they just spell out the acronyms for this figure on the Y axis? SAM = xxxxx; ZW3 = xxxxx; etc. There is room. Or, just say in caption "acronyms in a (new) Table X." One or a few of these options would make it much easier for the reader to appreciate such figures.

We've described axis labels and removed acronyms.

Also, in general it might be questioned how much of this is their work in this paper or comparing to prior analyses? Maybe they should specifically state " We analyzed...", or "This was analyzed...."

We followed the this advice in several places in the manuscript and supplement.

Supplement Figure 10. I know it is in the legend, but I would consider adding 'red' afteranthropogenic components (in red). And green after forcings on climate (in green). Would be easier for reader.

Done

Last, I do mention I am not an expert on the model, so cannot really evaluate the details of the model guts. However, it seems much of the background or context behind (at least earlier versions) the model has been published (reviewed already), so all seems fine.

REVIEWERS' COMMENTS:

Reviewer #1 (Remarks to the Author):

I am grateful to the authors for their very thorough work in responding to the reviews and for their revisions which have much improved the manuscript. I am now happy to recommend acceptance of this paper for publication.

Reviewer #2 (Remarks to the Author):

General Comment:

The revised manuscript is a significant improvement on the initial submission and the authors have done a very careful job at addressing the comments of the reviewers, which they should be congratulated for. The research presented is of interest and is now very well-articulated and I recommend publication of this manuscript. I only have some very minor comments for the authors that they may wish to consider should the research be accepted for publication, which I believe it should.

Minor comments:

Please note that line number is referred to as (L).

Main paper

1. L49: change 1990 to 1990s

2. L103: The authors may wish to consider the key finding by Conway and Cullen (2016) that the sensitivity of surface mass balance to changes in air temperature in the Southern Alps is greatly enhanced in overcast compared to clear-sky conditions, as controlled by more frequent melt and changes in precipitation phase leading to a strong albedo feedback. The research seems relevant in this context and describes many of the key processes described on L274-280.

Conway, J. P., & Cullen, N. J. (2016). Cloud effects on surface energy and mass balance in the ablation area of Brewster Glacier, New Zealand. *Cryosphere*, 10(1), 313-328. doi: 10.5194/tc-10-313-2016

3. L168: a period (of) mass loss

4. L228: repeat experiments (show) shows

5. L431: conditions (are) not well understood

Supplementary information

6. L597-604: It is excellent to see an effort made by the authors to introduce a subsurface thermal scheme that works for different surfaces. Given the effort, it might be of interest to provide a little more detail than what is provided on L601-604. The test result provided in Supplementary Figure 8 is difficult to see – consider changing the thickness or colour of the control to allow it to be

observed.

7. L603: increases (is) in generally

Reviewer #3 (Remarks to the Author):

Dear Nature Communications,

I looked at the revised manuscript and the point by point responses.

The authors have satisfactorily addressed my points. Some of my points were comprehensively addressed (even beyond what I had suggested) when they revised, to address other Reviewer comments (observations c/w modeling results being distinguished, e.g., new Fig 4).

And, they cleaned up the missing items I pointed out (e.g. in the legend on Fig. 1 and other figures).

I look forward to seeing the paper in print.

REVIEWERS' COMMENTS:

Reviewer #1 (Remarks to the Author):

I am grateful to the authors for their very thorough work in responding to the reviews and for their revisions which have much improved the manuscript. I am now happy to recommend acceptance of this paper for publication.

Reviewer #2 (Remarks to the Author):

General Comment:

The revised manuscript is a significant improvement on the initial submission and the authors have done a very careful job at addressing the comments of the reviewers, which they should be congratulated for. The research presented is of interest and is now very well-articulated and I recommend publication of this manuscript. I only have some very minor comments for the authors that they may wish to consider should the research be accepted for publication, which I believe it should.

Minor comments:

Please note that line number is referred to as (L).

Main paper

1. L49: change 1990 to 1990s

Done

2. L103: The authors may wish to consider the key finding by Conway and Cullen (2016) that the sensitivity of surface mass balance to changes in air temperature in the Southern Alps is greatly enhanced in overcast compared to clear-sky conditions, as controlled by more frequent melt and changes in precipitation phase leading to a strong albedo feedback. The research seems relevant in this context and describes many of the key processes described on L274-280.

Conway, J. P., & Cullen, N. J. (2016). Cloud effects on surface energy and mass balance in the ablation area of Brewster Glacier, New Zealand. *Cryosphere*, 10(1), 313-328. doi: 10.5194/tc-10-313-2016

We have cited this paper in the appropriate location.

3. L168: a period (of) mass loss

Done

4. L228: repeat experiments (show) shows

Done

5. L431: conditions (are) not well understood

Done

Supplementary information

6. L597-604: It is excellent to see an effort made by the authors to introduce a subsurface thermal scheme that works for different surfaces. Given the effort, it might be of interest to provide a little more detail than what is provided on L601-604. The test result provided in Supplementary Figure 8 is difficult to see – consider changing the thickness or colour of the control to allow it to be observed.

We have provided an updated version of this figure (now Supplementary Figure 4) based on our final model runs. The control line runs along the x axis because all anomalies are plotted relative to this control. This black line is clearly visible.

We've also provided an updated description of this result;

'We found that the more complete thermal calculations gave a very similar result to our simplified analysis. Including conductive melt under debris, and subsurface heat fluxes in snow and ice, the contribution of temperature variations to volume changes is generally decreased, while the contribution of precipitation variations to volume changes is close to zero over the full period. The contribution of other variables also changes, but remains minor.'

We feel that this is sufficient to make the point that the results (including or excluding the thermal calculations) are similar.

7. L603: increases (is) in generally

Done

Reviewer #3 (Remarks to the Author):

Dear Nature Communications,

I looked at the revised manuscript and the point by point responses.

The authors have satisfactorily addressed my points. Some of my points were comprehensively addressed (even beyond what I had suggested) when they revised, to address other Reviewer comments (observations c/w modeling results being distinguished, e.g., new Fig 4).

And, they cleaned up the missing items I pointed out (e.g. in the legend on Fig. 1 and other figures).

I look forward to seeing the paper in print.